# Geochemical Features of River Runoff and Their Effect on the State of the Aquatic Environment of Lake Onego

Natalia Kulik [1],*, Natalia Efremenko [1], Vera Strakhovenko [2], Natalia Belkina [1], Galina Borodulina [1], Ekaterina Gatalskaya [1], Viktor Malov [2] and Igor Tokarev [3]

[1] Northern Water Problems Institute of the Karelian Research Centre of the Russian Academy of Sciences, 185030 Petrozavodsk, Russia
[2] Sobolev Institute of Geology and Mineralogy Siberian Branch Russian Academy of Sciences, 630090 Novosibirsk, Russia
[3] Centre for Geo-Environmental Research and Modelling (GEOMODEL), Research Park, Saint-Petersburg State University, 198504 St. Petersburg, Russia
* Correspondence: nadiet11@yandex.ru

**Abstract:** This paper presents the results of seasonal observations of the geochemical composition of the waters of the large tributaries of Lake Onego. The mineralogy and geochemistry of the suspended matter and the isotopic composition (oxygen-18 and deuterium) of the river waters were studied for the first time. The dependence of the chemical and isotopic compositions of the tributary water on the season and characteristics of the catchment area (swampiness and lacustrine) was revealed. It is shown that the river waters belong to the bicarbonate class of the calcium group and have low mineralization, high color and a similar composition to the main minerals of the suspended matter. It is determined that the difference between the multielement spectra of the water and suspended matter of the different rivers is closely related to the geological and geomorphological structures of river basins. It is established that the quantitative characteristics of the mineral and organic parts of the suspended matter, the ratios of the different minerals and the size and patterning of the particles of detrital material in the tributaries differ. The change in the mineralogical and geochemical compositions of the suspended matter of each individual river over the year is insignificant. The influence of the river runoff on the formation of lake waters is manifested in the chemical composition of the lake waters. The quantitative ratios of the main ions, biogenic elements and microcomponents in lake water mainly correspond to their ratios in river waters. The mineral part of the dispersed sedimentary matter of the lake in its geochemical characteristics is close to the suspended matter of the river waters.

**Keywords:** Lake Onego; river runoff; isotopic composition of water; chemical composition of water; geochemical composition of suspended matter; mineralogical composition of suspended matter

## 1. Introduction

The study of the global, regional and local geochemical cycles of elements is one of the most relevant scientific orientations of the twenty-first century. Landscape-geochemical processes that determine the migration of elements and the forms of their presence in the environment determine their regional background levels and the formation of natural and man-made anomalies. Many authors point to the presence of an anthropogenic component of the geochemical background in almost all regions of the Earth [1–6]. Lakes, especially large ones (such as Lake Onego, the second largest reservoir in Europe), have a significant impact on the regional balance of elements. The special attention to large reservoirs is also due to the fact that they contain the main supply of surface freshwater that is actively used by humans for water supply and economic activities. A lake system is characterized by stability (i.e., stationarity), which largely depends on external factors. In a humid climate, river runoff is the main external factor affecting a lake. Therefore, observations of rivers are

an integral part of any research aimed at assessing the state of a lake system and its response to impacts of various kinds. Research based on an interdisciplinary systematic approach combines modern landscape-geographical, biogeochemical, mineralogical, hydrochemical and ecological methods [7–10].

The purpose of the presented work was to study the geochemical features of the river runoff into Lake Onego. Hydrochemical studies of the river runoff into Lake Onego were periodically conducted at the Northern Water Problems Institute of the Karelian Research Centre of the Russian Academy of Sciences (NWPI KarRC RAS) [11–14]. The main attention in these works is paid to the biogenic load on the lake, since eutrophication is one of the most acute problems of Lake Onego. The lake is a receiver of wastewater from three large industrial centers located on the shores of the northeastern bays: the cities of Petrozavodsk, Kondopoga and Medvezhyegorsk (Figure 1). Urban and industrial wastewater enriched with organic matter, nitrogen and phosphorus enhances the heterogeneity of the lake ecosystem (bays on the shores of which cities are located have a higher trophic status).

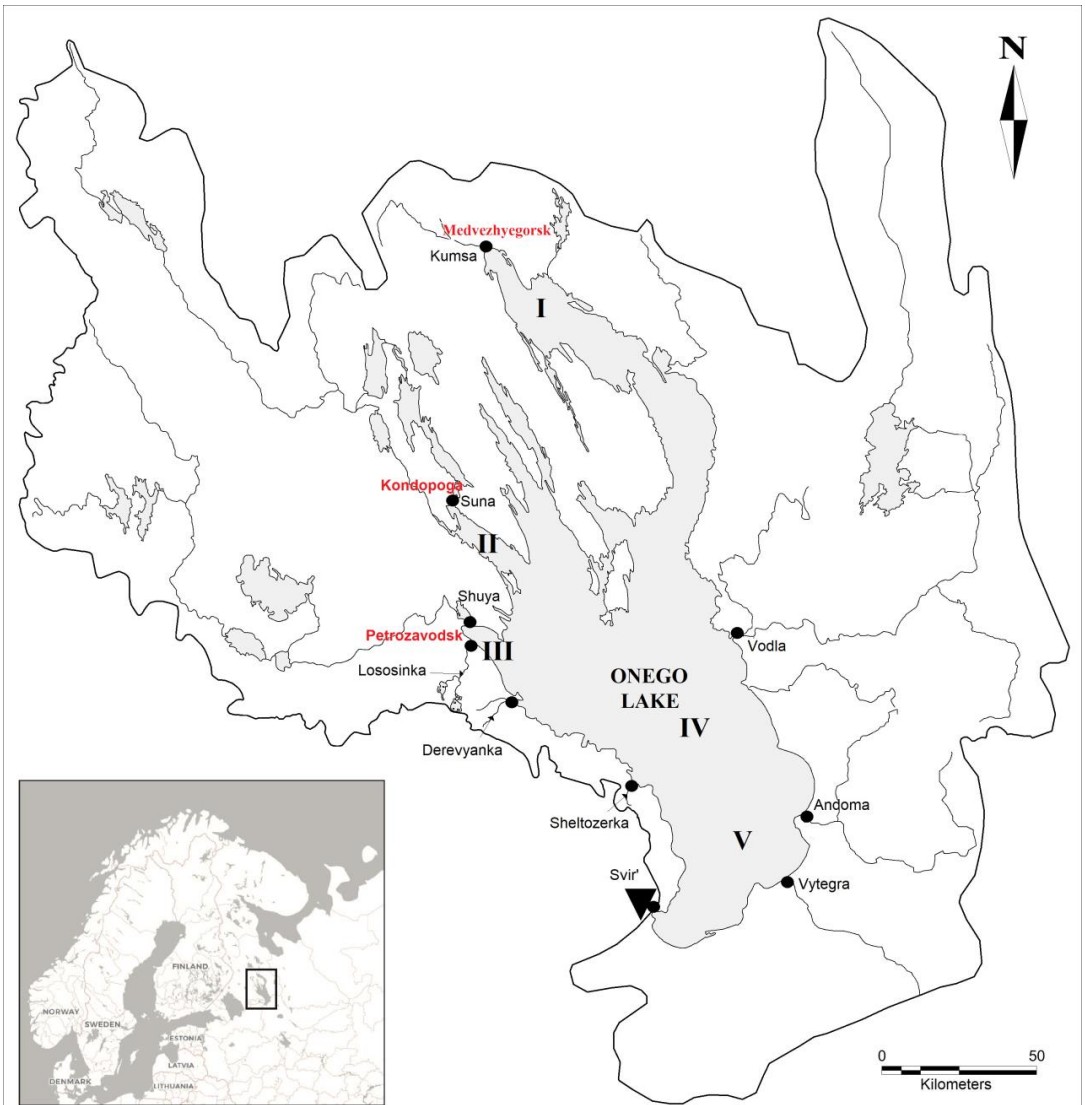

**Figure 1.** Study area: I—Gulf of Povenets; II—Kondopoga Bay; III—Petrozavodsk Bay; IV—Central Onego; V—Southern Onego.

Up to now, due attention has not been paid to the study of the trace element composition of river waters. Studies were carried out on individual rivers, according to a limited

list of components [15–17]. Studies of the material composition of the suspended part of the water runoff, which is given much attention in our article, have not been conducted. The objectives of our research included (1) a more detailed study of the geochemistry of waters (the list of components was expanded, the dissolved and suspended part of the substance was studied, as well as the mineralogical composition of the terrigenous material), (2) identification of the relationship of the geochemical composition of river waters with the geological features of the catchment area and (3) establishing the role of rivers in the formation of the trace element spectrum of lake waters.

## 2. Material and Methods

In 2020–2021, in all hydrological seasons, water was taken from the tributaries of Lake Onego, which comprise the main contribution to the river flow (Figure 1). Approximately 60% of the total river inflow into the lake accounts for the four large rivers, with a flow rate of more than 1 $km^3 \cdot year^{-1}$—Vodla, Shuya, Suna and Andoma rivers. The rivers with an increased flow rate include the Vytegra (0.52 $km^3 \cdot year^{-1}$), the other tributaries have a flow rate of less than 0.5 $km^3 \cdot year^{-1}$. The average annual water flow of the Svir' river at its source is 18.6 $km^3$. Table 1 provides, in brief, the characteristics of the rivers studied.

**Table 1.** Characteristics of the tributaries and the source of Lake Onego and their catchments [14,18].

| Geological Structure | Coast | River | Flow Volume, km³/Year | Length, km | Watershed | | |
| --- | --- | --- | --- | --- | --- | --- | --- |
| | | | | | Area, Thousand km² | Lacustrine Nature of the Territory, % | Swampiness of the Territory, % |
| Fennoscandian Crystalline Shield (FCS) (I) | Northwest | Lososinka | 0.12 | 25 | 0.3 | 5.7 | 10 |
| | | Shuya | 3.09 | 279 | 10.3 | 10.4 | ~20 |
| | | Suna | 2.34 | 282 | 7.67 | 12.5 | 19 |
| | North | Kumsa | 0.23 | 67 | 0.74 | 8.5 | 7 |
| | Eastern | Vodla | 4.63 | 406 | 13.7 | 5.6 | 24 |
| | Southeastern | Andoma | 1.03 | 142 | 2.57 | 1.3 | 12 |
| | Southwest | Derevyanka | 0.03 | 20 | 0.093 | 0.4 | – |
| | | Sheltozerka | 0.03 | 11 | 0.069 | 3.9 | – |
| The central part of the East European Platform (II) | Southern | Vytegra | 0.52 | 40 | 1.67 | <1 | 12 |
| | | Svir' | 24.9 | 224 | 84.4 | – | – |

«–»—No data.

The objects of study weregrouped according to the geographical location of the catchments, taking into account the geological and geomorphological structures of the lake basin. Thus, the investigated rivers, Lososinka, Shuya, Suna (northwest coast), Kumsa (north coast), Vodla (east coast), Andoma (southeast coast) and Derevyanka and Sheltozerka (southwest coast), drain the territory of the Archean–Proterozoic Fennoscandian Crystalline Shield (FCS), folded atonalite–trondyemiteatonalite–trondyemite gneisses, granites, migmatites and granulites, tholeiitic and ferrometabasalts, metadacites, metalparitesand conglomerates, as well as gabbro–anorthositesgabbro–anorthosites and alkaline granites formed from 3240 to 2680 million years [19]. The VytegraRiver (southern coast) drains the central part of the East European Platform and is composed of Vend–Paleozoic rocks of the platform cover. The plate complex includes terrigenous formations of the Middle and Upper Devonian, Carboniferous–Lower Permian carbonate–sulfate rocks, red flowers of the Upper Permian and Triassic, and low-power continental marine sediments of the Jurassic, Cretaceous and anthropogenic [18,19]. In our opinion, a powerful cover of Quaternary sediments (interglacial, continental and marine formations of the early, middle and late Pleistocene [20–23]) formed in separate areas of the eastern and western coasts, playinga major role in the formation of the chemical composition of river waters (Figure 1).

The Svir' River is the source of Lake Onego, the chemical composition of the water of which is formed by the waters of Southern Onego.

Water sampling was carried out from the surface horizon into ten-liter polyethylene canisters. In laboratory conditions, the suspended matter was divided into fractions by sequential filtration through membrane filters with different pore diameters according to ISO 11923:1997. The dissolved forms of the elements were studied in the filtrate obtained after passing the source water through a filter of $\varnothing$ 0.45 μm. When the source water was successively passed through filters of $\varnothing$ 0.8 and $\varnothing$ 0.45 μm, the fine fraction of the suspended substance was estimated by the difference in the content of the microcomponents on the filtrates. The suspended matter on the filters $\varnothing$ 0.8 μm was considered as a coarse fraction of the suspended matter in river water. Filters with suspended matter were dried in a drying cabinet at 105 °C to a constant mass. The solid precipitate was converted to a dissolved state for ICP analysis in an acidic medium (in 10 mL of concentrated nitric acid) in the Berghof SpeedWave®four microwave system.

The material composition of the suspended matter isolated on the filters from the water of the tributaries of Lake Onego was studied by optical and scanning electron microscopy (SEM) on a MIRA 3 TESCAN (Czech Republic) with an INCA Energy microanalyzer (microprobe) (Oxford Instruments, UK) at the analytical Centre of the IGM SB RAS (Novosibirsk).

The analysis of the chemical composition of water (pH, $NH_4^+NH$, $NO_3^-NO^-$, $N_{tot}$, $P_{min}$, $P_{tot}$, BOD, permanganate index (PI), COD, Si and color) was carried out at the NWPI KarRC RAS (Petrozavodsk) in accordance with the methods generally accepted in hydrochemical research [24–26]. The microcomponent composition in water and suspended matter was analyzed in the NWPI KarRC RAS by atomic absorption (AA6800 spectrometer, Shimadzu, Japan) and ICP-MS (7500a spectrometer, Agilent Technologies, USA) methods on the scientific equipment of the Core Facility of the Karelian Research Centre of the Russian Academy of Sciences (Petrozavodsk) and at the analytical Centre of the IGM SB RAS (Novosibirsk).

Measurements of the deuterium and oxygen-18 contents in the water were carried out on a Picarro L-2120-i laser infrared analyzer of the isotopic composition of water (Centre for Geo-Environmental Research and Modelling (GEOMODEL), Research Park, Saint-Petersburg State University). All results are providedin ppm relative to the composition of average oceanic water (SMOW). The IAEA standards V-SMOW-2, GISP and SLAP, as well as the standards of the American Geological Society, were used as comparison samples: USGS-45 and USGS-46. The measurement uncertainty is ±0.1% for oxygen-18 and ±1% for deuterium.

## 3. Results and Discussion

### 3.1. Chemical Composition of River Water

The results of the study of the chemical composition of the waters showed that, despite the similarity of the main chemical characteristics due to the common climatic conditions of the formation of river runoff, due to the heterogeneity of the geological and geomorphological structures of the basin and its hydrographic features, the material compositions of the river waters within the region differ.

For the rivers with a catchment area located on the Archean–Proterozoic Fennoscandian Crystalline Shield (FCS), during the observation period, the water mineralization varied from 11 to 60 mg/L (average annual value of 29 mg/L). In the Vytegra River, which drains carboniferous limestones, the water mineralization was much higher and ranged from 110 to 193 mg/L (on average 161 mg/L) (Table 2). For all tributaries of the lake, with the exception of the Suna River and for the source of the lake (i.e., the Svir' River), the mineralization in the summer and winter seasons was 1.5–2 times higher than in spring and autumn. In the water of the Suna River, the maximum mineralization was noted only in winter. The mineralization of the water at the source of the lake did not change during the entire observation period.

According to Alekin's classification [27], the waters of all the rivers studied are of the bicarbonate–calcium type (Table 3). The ratio of Ca and Mg ions in water is stable

(Ca:Mg = 1.5). The concentration of alkali metals in water is insignificant (the average content of $Na^+$Na = 2.3 mg/L and $K^+$ = 0.70 mg/L). The content of chloride ions in river waters is small (ranges from 4 to 24%-eq) and generally exceeds the content of sulfate ions (from 2 to 16%-eq), with the exception of the water of the Suna and Derevyanka rivers in the autumn, and the Svir' River throughout the year. Increased concentrations of $Cl^-$Cl with a simultaneous increase in the $Na^+$Na content were observed during the autumn flood in the Lososinka River, and a significant part of its catchment area is located on the territory of the City of Petrozavodsk. In the anionic composition, attention should also be paid to the content of anions of organic acids. Their increased content was observed in rivers with a heavy swampy catchment (Shuya, Sheltozerka, Derevyanka and Andoma). Thus, in the Shuya River (swampiness of territory $\approx$ 20%) in the spring and autumn periods, the proportion of organic ions reached 34%-eq;in the southern tributaries, it varied from 23 to 36%-eq (Table 3).

The contents of organic matter (OM) and biogenic elements (BEs) in tributaries depend on a number of factors, among which the main role is played by the climate and the geological and geomorphological features of the region, as well as economic activity in the catchment area. Like most rivers of the taiga zone, the studied tributaries are enriched with humus substances. The maximum values of OM indicators in rivers were observed in spring and autumn. It is during the spring and autumn that the maximum intake of allochthonous OM into the lake occurs, affecting the migration of metals [16,28]. The following features of the rivers' OM are noted. The values of the OM content for the two largest tributaries (i.e., Shuya and Vodla) were comparable and close to each other. In the waters of the Suna River, the values of PI, COD and color were weakly subject to seasonal changes during the year due to the high regulation of water runoff. There were no seasonal changes in these indicators during the observation period for the Kumsa River. In the water of small tributaries located on the border of the FCS (Derevyanka and Sheltozerka), the greatest seasonal fluctuations of the indirect indicators of OM were noted. The maximum content of OM in them was recorded during the rainy period (summer–autumn). The water color, PI and COD in these rivers were 1.5–2 times higher than in the spring and winter. High ratios of PI/COD = 47%, color/PI = 6.3 and color/COD = 3.0 with a relatively low value of BOD (average = 2.0) indicate the predominance of organic substances of humus origin resistant to the biochemical oxidation formed in soils.

The average annual concentrations of total phosphorus ($P_{tot}$) ranged from 35 to 168 μg/L and for mineral phosphorus ($P_{min}$) from 9 to 58 μg/L (Table 2). During the year, the maximum $P_{tot}$ content of all of the rivers studied was noted in the summer and winter. The minimum concentrations of $P_{min}$ (approximately1 μg/L) and $P_{tot}$ (6–24μg/L) during the observation period were established for the Suna and Kumsa rivers, whose catchments are characterized by a relatively high degree of lacustrine and swampiness. For the Shuya and Lososinka rivers that drain the territory of the City of Petrozavodsk, the content of $P_{tot}$ ranged from 30 to 103 μg/L. The share of $P_{min}$ from $P_{tot}$ was 30–59%. The average annual contents of $P_{min}$ and $P_{tot}$ in the waters of the Vodla, Andoma and Vytegra rivers were 10 and 48 μgP/L, respectively. The upper limit of their content reached 16 ($P_{min}$) and 62 ($P_{tot}$) μgP/L. The highest concentrations of P were obtained in the waters of the rivers of the southwestern coast (Sheltozerka and Derevyanka).

**Table 2.** Chemical composition of the water of the tributaries of Lake Onego («–»—no data).

| Geological Structure | Coast | River | Season | pH | NH$_4^+$ mgN/L | NO$_3^-$ mgN/L | Ntot mgN/L | Norg mgN/L | Pmin µg/L | Ptot µg/L | BOD mgO/L | PI, mgO/L Not Filtered | COD, mgO/L Filtered | COD, mgO/L Not Filtered | Si (Filtered) mg/L | Color °Pt | Σions mg/L |
|---|---|---|---|---|---|---|---|---|---|---|---|---|---|---|---|---|---|
| The Fennoscandian Crystalline Shield (FCS) (I) | Northwest | Lososinka (before the city) | Spring | 7.01 | 0.013 | <0.01 | 0.40 | 0.39 | 17 | 46 | 1.91 | 18.8 | 27.9 | 29.4 | 2.67 | 136 | 26 |
| | | | Summer | 7.37 | 0.051 | 0.009 | 0.60 | 0.54 | 48 | 91 | 1.74 | 19.9 | 43.4 | 45.4 | 2.68 | 154 | 39 |
| | | | Autumn | 7.01 | 0.046 | 0.042 | 0.54 | 0.45 | 26 | 50 | 1.57 | 23.5 | 44.8 | 45.6 | 2.74 | 169 | 30 |
| | | | Winter | 6.59 | 0.055 | 0.104 | 0.67 | 0.51 | 15 | 155 | 2.67 | 26.3 | 53.9 | 62.5 | 3.46 | 170 | 22 |
| | | Lososinka (the mouth of the river) | Spring | 7.14 | 0.024 | 0.017 | 0.43 | 0.39 | 22 | 53 | 2.01 | 18.8 | 30.8 | 33.5 | 2.7 | 135 | 32 |
| | | | Summer | 7.29 | 0.108 | 0.093 | 0.66 | 0.46 | 61 | 103 | 2.04 | 18.3 | 41.9 | 46.2 | 2.64 | 137 | 51 |
| | | | Autumn | 7.26 | 0.076 | 0.072 | 0.68 | 0.53 | 38 | 69 | 2.12 | 22.7 | 45.2 | 48.8 | 2.52 | 169 | 38 |
| | | | Winter | 7.28 | 0.070 | 0.136 | 0.46 | 0.25 | 18 | 62 | 1.98 | 14.6 | 35.3 | 36.5 | 3.05 | 117 | 60 |
| | | Shuya | Spring | 6.39 | 0.013 | 0.029 | 0.38 | 0.34 | 4 | 31 | 2.31 | 19.6 | 36.8 | 38.7 | 2.70 | 152 | 15 |
| | | | Summer | 7.01 | 0.032 | 0.009 | 0.43 | 0.39 | 9.8 | 45 | 1.36 | 12.6 | 27.5 | 31.0 | 1.22 | 78 | 23 |
| | | | Autumn | 6.58 | 0.041 | 0.038 | 0.54 | 0.46 | 14 | 36 | 1.63 | 22.7 | 42.5 | 48.8 | 2.2 | 153 | 18 |
| | | | Winter | 6.06 | 0.038 | 0.182 | 0.82 | 0.60 | 8 | 53 | 1.82 | 20.2 | 45.7 | 47.7 | 3.39 | | 27 |
| | | Suna | Spring | 7.13 | 0.009 | 0.055 | 0.35 | 0.29 | 1 | 11 | 1.49 | 10.0 | 21.6 | 24.6 | 1.94 | 60 | 17 |
| | | | Summer | 6.70 | 0.024 | 0.041 | 0.35 | 0.29 | 1 | 12 | 0.74 | 10.9 | 19.9 | 20.7 | 2.02 | 62 | 16 |
| | | | Autumn | 6.91 | 0.024 | 0.082 | 0.30 | 0.19 | 1 | 1 | 1.68 | 10.1 | 19.1 | 21.9 | 2.09 | 61 | 11 |
| | | | Winter | 6.75 | 0.045 | 0.089 | 0.33 | 0.20 | 0 | 1 | 0.55 | 11 | 27.7 | 25.9 | 2.12 | 77 | 14 |
| | North | Kumsa | Spring | 7.19 | 0.010 | 0.010 | 0.34 | 0.32 | 1 | 9 | 2.22 | 12.6 | 25.5 | 25.5 | 2.10 | 68 | 30 |
| | | | Summer | 7.30 | 0.019 | 0.021 | 0.38 | 0.34 | 0.3 | 12 | 1.34 | 10.7 | 25.1 | 30.2 | 2.00 | 59 | 40 |
| | | | Autumn | 7.09 | 0.046 | 0.038 | 0.41 | 0.33 | 1 | 10 | 1.07 | 11.8 | 29.1 | 31.2 | 2.45 | 75 | 37 |
| | | | Winter | 7.22 | 0.047 | 0.065 | 0.46 | 0.35 | 0 | 24 | 1.81 | 12.2 | 28.2 | 30.9 | 2.7 | 65 | 42 |
| | Eastern | Vodla | Spring | 6.83 | 0.010 | <0.01 | 0.37 | 0.36 | 9 | 45 | 2.48 | 18.8 | 36.8 | 38.9 | 2.15 | 120 | 21 |
| | | | Summer | 6.33 | 0.030 | 0.006 | 0.47 | 0.43 | 2.3 | 43 | 2.86 | 12.6 | 23.3 | 29.5 | 0.84 | 71 | 30 |
| | | | Autumn | 6.67 | 0.066 | 0.026 | 0.68 | 0.59 | 8 | 34 | 3.65 | 24.7 | 48.6 | 52 | 2.08 | 149 | 27 |
| | | | Winter | 6.83 | 0.090 | 0.132 | 0.78 | 0.56 | 10 | 44 | 1.61 | 16.9 | 3.85 | 40.2 | 2.43 | 92 | 39 |
| | Southeastern | Andoma | Spring | 6.89 | 0.016 | 0.041 | 0.51 | 0.45 | 11 | 33 | 1.65 | 18.3 | 29.0 | 37.9 | 1.62 | 140 | 42 |
| | | | Summer | 6.99 | 0.025 | 0.026 | 0.85 | 0.80 | 11 | 62 | 2.98 | 26.4 | 46.0 | 54.5 | 2.06 | 196 | 57 |
| | | | Autumn | 6.16 | 0.050 | 0.019 | 0.97 | 0.90 | 14 | 44 | 2.84 | 34.3 | 67.4 | 70.6 | 2.91 | 215 | 19 |
| | | | Winter | 6.91 | 0.097 | 0.163 | 0.84 | 0.58 | 8 | 61 | 3.45 | 17.7 | 39.5 | 40.9 | 2.97 | 127 | 33 |
| | Southwest | Derevyanka | Spring | 7.20 | 0.010 | 0.043 | 0.55 | 0.50 | 25 | 67 | 3.10 | 29.3 | 45.7 | 48.9 | 3.09 | 208 | 44 |
| | | | Summer | 7.21 | 0.021 | 0.076 | 1.24 | 1.14 | 58 | 127 | 2.84 | 54.2 | 78.0 | 94.7 | 4.73 | 358 | 53 |
| | | | Autumn | 7.21 | 0.043 | 0.104 | 1.13 | 0.98 | 37 | 78 | 2.36 | 45 | 72.6 | 751 | 2.09 | 271 | 53 |
| | | | Winter | 7.11 | 0.218 | 0.667 | 1.55 | 0.66 | 36 | 168 | 2.84 | 14.1 | 37.2 | 423 | 3.56 | 107 | 92 |
| | | Sheltozerka | Spring | 6.89 | 0.010 | <0.01 | 0.58 | 0.57 | 9 | 39 | 1.96 | 22.0 | 42.8 | 46.3 | 2.30 | 160 | 23 |
| | | | Summer | 7.21 | 0.027 | 0.088 | 1.11 | 1.00 | 47 | 89 | – | 26.4 | 50.1 | 51.7 | 3.73 | 211 | 75 |
| | | | Autumn | 6.65 | 0.025 | 0.028 | 0.93 | 0.88 | 11 | 36 | 7.3 | 33.8 | 65.7 | 70.4 | 3.1 | 213 | 26 |
| | | | Winter | 6.82 | 0.103 | 0.508 | 1.33 | 0.74 | 26 | 76 | 2.21 | 14.5 | 46.3 | 50.2 | 4.05 | 151 | 48 |

**Table 2.** *Cont.*

| Geological Structure | Coast | River | Season | pH | $NH_4^+$ mgN/L | $NO_3^-$ mgN/L | Ntot mgN/L | Norg mgN/L | Pmin µg/L | Ptot µg/L | BOD mgO/L | PI, mgO/L Not Filtered | COD, mgO/L Filtered | COD, mgO/L Not Filtered | Si (Filtered) mg/L | Color °Pt | Σions mg/L |
|---|---|---|---|---|---|---|---|---|---|---|---|---|---|---|---|---|---|
| The central part of theEast European Platform (II) | Southern | Vytegra | Spring | 7.71 | 0.025 | 0.076 | 0.35 | 0.25 | 6 | 25 | 1.43 | 13.4 | 25.3 | 30.1 | 1.71 | 89 | 110 |
| | | | Summer | 8.09 | 0.029 | 0.013 | 0.54 | 0.50 | 16 | 59 | 1.63 | 6.9 | 15.5 | 17.9 | 1.78 | 38 | 185 |
| | | | Autumn | 7.87 | 0.109 | 0.061 | 0.76 | 0.59 | 14 | 45 | 2.91 | 16.4 | 32.6 | 34.7 | 2.31 | 102 | 156 |
| | | | Winter | 7.54 | 0.079 | 0.215 | 0.80 | 0.50 | 15 | 61 | – | 17.7 | 24.8 | 26.1 | 3.38 | 59 | 193 |
| | | Svir' (the source of the lake) | Spring | 7.40 | 0.009 | 0.121 | 0.28 | 0.15 | 11.3 | 14 | 0.94 | 6.4 | 19.3 | 20.5 | 0.21 | 28 | 40 |
| | | | Summer | 7.27 | 0.017 | 0.119 | 0.41 | 0.27 | 0.7 | 10 | 0.44 | 7.8 | 19.1 | 19.6 | 0.14 | 28 | 38 |
| | | | Autumn | 7.47 | 0.032 | 0.128 | 0.45 | 0.29 | 0.7 | 9 | 0.45 | 8.2 | 18.5 | 19.7 | 0.28 | 33 | 37 |
| | | | Winter | 7.37 | 0.035 | 0.164 | 0.39 | 0.19 | 0.7 | 18 | 0.62 | 6.8 | 18.4 | 20.0 | 0.34 | 34 | 38 |

**Table 3.** Chemical composition of the river waters.

| Geological Structure | Coast | River | Spring | Summer | Autumn | Winter |
|---|---|---|---|---|---|---|
| I | Northwest | Lososinka (before the city) | $\frac{Ca40Mg37Na18K2NH_4\,2}{HCO_3\,70\,A_{org}12Cl9SO_4\,9}$ | $\frac{Ca44Mg38Na15K2NH_4\,1}{HCO_3\,76\,A_{org}10Cl7SO_4\,7}$ | $\frac{Ca42Mg37Na18K3NH_4\,1}{HCO_3\,64\,A_{org}20Cl8SO_4\,8}$ | $\frac{Ca42Mg37Na18K3NH_4\,1}{HCO_3\,64\,A_{org}20Cl8SO_4\,8}$ |
| | | Lososinka (the mouth of the river) | $\frac{Ca44Mg34Na17K2NH_4\,3}{HCO_3\,47\,A_{org}36Cl9SO_4\,8}$ | $\frac{Ca43Mg35Na18K2NH_4\,1}{HCO_3\,71\,A_{org}9Cl9SO_4\,9NO_3\,2}$ | $\frac{Ca41Mg35Na21K3NH_4\,1}{HCO_3\,64\,A_{org}14Cl13SO_4\,8NO_3\,1}$ | $\frac{Ca38Na34Mg26K2}{HCO_3\,53\,Cl25A_{org}16SO_4\,6}$ |
| | | Shuya | $\frac{Ca45Mg26Na22K4NH_4\,3}{HCO_3\,41\,A_{org}36Cl13SO_4\,10}$ | $\frac{Ca39Mg29Na27K4NH_4\,1}{HCO_3\,65\,A_{org}12Cl12SO_4\,11}$ | $\frac{Ca43Mg29Na24K4NH_4\,1}{HCO_3\,45\,A_{org}34Cl12SO_4\,8NO_3\,1}$ | $\frac{Ca37Mg29Na29K4}{HCO_3\,41A_{org}27Cl21SO_4\,10NO_3\,1}$ |
| | | Suna | $\frac{Ca46Mg29Na18K5NH_4\,2}{HCO_3\,61\,A_{org}19SO_4\,11Cl7NO_3\,2}$ | $\frac{Ca46Mg29Na20K5NH_4\,1}{HCO_3\,61SO_4\,15A_{org}14Cl8NO_3\,2}$ | $\frac{Ca47Mg29Na20K4NH_4\,1}{A_{org}49HCO_3\,27\,SO_4\,14Cl8NO_3\,2}$ | $\frac{Ca47Mg28Na20K4NH_4\,1}{HCO_3\,53A_{org}33Cl7SO_4\,6NO_3\,1}$ |
| | North | Kumsa | $\frac{Ca46Mg35Na15K2NH_4\,2}{HCO_3\,77Cl11SO_4\,7\,A_{org}5}$ | $\frac{Ca48Mg35Na14K2}{HCO_3\,82Cl9SO_4\,6A_{org}3}$ | $\frac{Ca48Mg33Na16K2NH_4\,1}{HCO_3\,75\,A_{org}10Cl10SO_4\,5}$ | $\frac{Ca48Mg33Na16K2NH_4\,1}{HCO_3\,74Cl13\,A_{org}8SO_4\,5}$ |
| | Eastern | Vodla | $\frac{Ca44Mg38Na14K2NH_4\,2}{HCO_3\,65\,A_{org}21Cl7SO_4\,7}$ | $\frac{Ca46Mg37Na14K2}{HCO_3\,79\,A_{org}10Cl6SO_4\,5}$ | $\frac{Ca46Mg36Na14K3NH_4\,1}{HCO_3\,65\,A_{org}23Cl7SO_4\,5}$ | $\frac{Ca47Mg35Na15K2NH_4\,1}{HCO_3\,81\,A_{org}9Cl7SO_4\,3}$ |
| | Southeastern | Andoma | $\frac{Ca58Mg29Na8K3NH_4\,2}{HCO_3\,93\,SO_4\,3Cl2A_{org}1NO_3\,1}$ | – | $\frac{Ca43Mg32Na17K7NH_4\,1}{HCO_3\,51\,A_{org}34Cl10SO_4\,5}$ | $\frac{Ca52Mg34Na11K3NH_4\,1}{A_{org}51HCO_3\,41\,Cl5SO_4\,3}$ |
| | Southwest | Derevyanka | $\frac{Ca45Mg34Na16K3NH_4\,1}{HCO_3\,59\,A_{org}23SO_4\,11Cl7}$ | $\frac{Ca43Mg35Na18K4}{HCO_3\,58\,A_{org}25SO_4\,11Cl5NO_3\,1}$ | $\frac{Ca41Mg36Na19K4}{HCO_3\,59\,A_{org}20SO_4\,12Cl8NO_3\,1}$ | $\frac{Ca41Mg35Na18K6}{HCO_3\,65\,SO_4\,16A_{org}10Cl8NO_3\,1}$ |
| | | Sheltozerka | $\frac{Ca49Mg32Na15K3NH_4\,2}{HCO_3\,51\,A_{org}36SO_4\,7Cl6}$ | $\frac{Ca49Mg36Na12K3}{HCO_3\,79A_{org}11SO_4\,5Cl4NO_3\,1}$ | $\frac{Ca47Mg34Na14K4}{HCO_3\,54\,A_{org}32Cl7SO_4\,6NO_3\,1}$ | $\frac{Ca45Mg33Na15K7NH_4\,1}{HCO_3\,66\,A_{org}19Cl7SO_4\,7NO_3\,1}$ |

**Table 3.** *Cont.*

| Geological Structure | Coast | River | Spring | Summer | Autumn | Winter |
|---|---|---|---|---|---|---|
| II | Southern | Vytegra | $\dfrac{\text{Ca67Mg26Na5K1NH}_4\ 1}{\text{HCO}_3\ 83\ A_{org}12\text{Cl3SO}_42}$ | – | $\dfrac{\text{Ca65Mg28Na6K1}}{\text{HCO}_3\ 83\ A_{org}12\text{Cl3SO}_42}$ | $\dfrac{\text{Ca64Mg26Na9K1}}{\text{HCO}_3\ 84\text{Cl7}A_{org}6\text{SO}_43}$ |
| | | Svir' | $\dfrac{\text{Ca49Mg32Na15K3NH}_4\ 1}{\text{HCO}_3\ 71\ \text{SO}_414\text{Cl9}A_{org}6}$ | $\dfrac{\text{Ca48Mg32Na17K3}}{\text{HCO}_3\ 74\ \text{SO}_415\text{Cl8}A_{org}3}$ | $\dfrac{\text{Ca48Mg32Na17K3}}{\text{HCO}_3\ 70\text{SO}_415\text{Cl8}\ A_{org}5\text{NO}_32}$ | $\dfrac{\text{Ca47Mg34Na16K3}}{\text{HCO}_3\ 70\ \text{SO}_415\text{Cl8}A_{org}7}$ |

«–»—The obtained result is not given due to the fact of abnormal valuesfor$HCO_3$ and $A_{org}$.

Of the forms of N in the waters of the tributaries, the $N_{org}$ form prevailed. The concentration of $N_{org}$ in the rivers draining FCS rocks varied from 0.19 to 0.60 mgN/L (on average 0.36 mgN/L) and for the rivers of the east coast and the Vytegra River from 0.45 to 0.90 mgN/L (on average 0.66 mgN/L). The greatest fluctuations in the concentrations and the maximum content of $N_{org}$ were obtained in the Sheltozerka and Derevyanka rivers, from 0.5 to 1.1 mgN/L (on average 0.84 mgN/L). Among the mineral forms of N in the river runoff, the $NO_3^-NO$ form prevailed, and its increased concentrations were especially observed in the winter. The lowest concentrations were observed for the rivers of the northern and northwestern coasts, the average values for the rivers of the eastern and southern coasts, and the maximum values for the rivers of the southwestern coast. The $NH_4^+NH$ form of N prevailed in the Lososinka, Shuya, Kumsa, Vodla and Vytegra rivers during the open water period (Table 2).

Si enters river waters as a result of the active leaching of rocks in the catchment area of Lake Onego. In the surface waters of Karelia, its main dissolved form is silicic (orthosilicic) acid, which is most easily absorbed by diatoms. Its content usually does not exceed 6 mgSi/L (on average 1.8 mgSi/L), which is determined by the low solubility of $SiO_2$ in water. The concentration of the suspended forms of Si periodically exceeds the concentration of the dissolved forms [29]. Dissolved forms of Si are also represented by polysilic acids, having a variable composition of $xSiO_2 \cdot yH_2O$ and Si-containing organic compounds (mainly as part of complexes with humic substances). According to [30], in the summer period in the Shuya River, where the concentration of dissolved Si was 2.1 mgSi/L, the proportion of polysilic acids was 5%, and the proportion of organic Si was 28%. The average content of dissolved Si in the river runoff in our studies was 1.9 mgSi/L. Its maximum concentrations were typical for the winter period (on average 3.1 mgSi/L). The lowest concentrations were observed in the summer period. The exceptions were the Sheltozerka and Derevyanka rivers, in whose water the Si content increased 1.5 times in the summer compared to spring. Such an increase may be due to the acceleration of denudation processes in quartzite sandstones with an increase in the temperature and the active removal of material in the summer from open pits located in the catchment area of these rivers.

The Svir' River is the source of Lake Onego, the chemical composition of the water of which is determined by the lake regime. The waters of the river, such as the waters of all tributaries, belong to the bicarbonate class of the calcium group. The concentration of hydrocarbonates during the entire observation period was approximately 70%-eq. In the anionic composition, sulfate ions (on average 14.6%-eq) prevailed over chloride ions (on average 8.35%-eq). The contents of OM and BE in the waters of the river were lower than in the studied tributaries and were mainly determined by the flow of internal processes in Lake Onego [11]. There were no significant seasonal fluctuations in the chemical composition of the river water by components over the year.

The influence of the river runoff on the formation of the chemical composition of the water in the Lake Onego is also confirmed by the isotopic data. The contents of the deuterium and oxygen-18 in the water of the tributaries ($\delta^{18}O = -14.4..-9.1‰$ and $\delta^2H = -102..-73‰$) variedin a wider range than in the lake ($\delta^{18}O = -11.5..-9.3‰$ and $\delta^2H = -85..-71‰$). On the isotope diagram (Figure 2), most of the figurative points for the water of the tributaries are shifted to the rightfrom the local meteoric water line (LMWL) due to the fact of evaporation and have the less depleted composition by heavy isotopes ("heavier" isotope composition). For 2012–2018, the weighted mean of the isotope composition of the precipitation was$\delta^{18}O = -11.9‰$ and $\delta^2H = -85‰$, and the equation of the LMWL is $\delta^2H = 7.7 \times \delta^{18}O + 7$ [31], which differs slightly from the global meteoric water line.

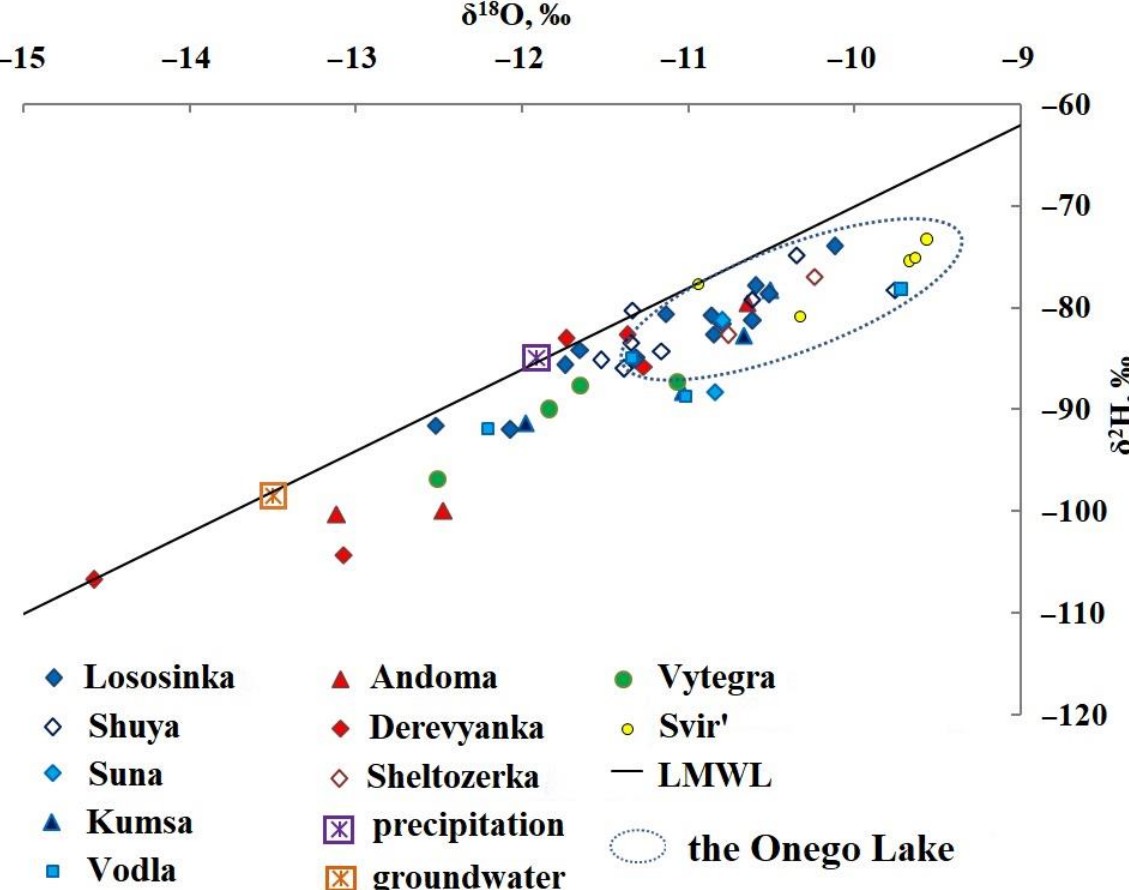

**Figure 2.** The isotopic composition of the water in the tributaries of Lake Onego.The oval is the main part (95%) of the water samples of Lake Onego; the precipitation is the weighted mean of the isotope composition; and the groundwater is the arithmetic average of756 samples (approximately87% from the FCS and the remaining from the EEP area).

There is a clear differentiation of the water isotopic composition for most of the tributaries by season (Table 4). In the summer and autumn, the content of heavy isotopes in river water is significantly higher than in winter due to the fact of evaporation and the influence of summer precipitation [32]. The evaporation effect is unambiguously indicated by the deuterium excess, which was less on average (d $^2$Hexcess = 4.0) for the river water to the free term of the LMWL equation by magnitude. The isotopic composition of the water of the Svir' River, which outflows from Lake Onego, was logically close to the isotopic composition of the lake water, and in summer, the water of the Svir' River was the most enriched with heavy isotopes among all of the rivers (Figure 2, Table 4).

In winter, the isotope composition of the water in the tributaries was sometimes more isotopically depleted than the weighted annual precipitation due to the transition of the rivers to supplying groundwater. The small rivers Andoma and Derevyanka have a predominant groundwater source throughout the whole year, in contrast to the rivers outflowing from lakes. Thisis evidenced by the light isotopic composition of the water even in the summer period: $\delta^{18}O < -12‰$ and $\delta^2H < -95‰$. The lightest isotopic composition of $\delta^{18}O = -14.6‰$ and $\delta^2H = -107‰$ was observed in the Derevyanka River for the end of the winter low flow in March during the thaw. At the same time, evaporation still continues to impact on these rivers, as the groundwater temperature is above the freezing point, and during some parts of winter, the rivers remain uncovered with ice, or the river has polynyas due to the fast current. So, the deuterium excess for the river water, on average (d 2Hexcess = 3.8), continues to be less than the free term of the LMWL equation.

**Table 4.** Isotopic composition of the river waters.

| Geological Structure | Coast | River | n | Range | Winter–Spring | | | Summer–Autumn | | |
|---|---|---|---|---|---|---|---|---|---|---|
| | | | | | $\delta^{18}$O, ‰ | $\delta^2$H, ‰ | d ($^2$HExcess), ‰ | $\delta^{18}$O, ‰ | $\delta^2$H, ‰ | d ($^2$HExcess), ‰ |
| Fennoscandian Crystalline Shield (FCS)(I) | Northwest | Lososinka | 15 | Min. | −12.5 | −92 | 8.0 | −11.3 | −85 | 5.4 |
| | | | | Max. | −10.9 | −93 | −5.8 | −10.1 | −74 | 6.8 |
| | | Shuya | 12 | Min. | −11.5 | −85 | 7.0 | −9.8 | −78 | 0.4 |
| | | | | Max. | −10.4 | −75 | 8.2 | | | |
| | | Suna | 2 | Min. | − | − | − | −10.8 | −88 | −1.6 |
| | | | | Max. | − | − | − | −10.8 | −81 | 5.4 |
| | North | Kumsa | 4 | Min. | −12.0 | −91 | 5.0 | −10.7 | −83 | 2.6 |
| | | | | Max. | −11.0 | −88 | 0.0 | −10.5 | −78 | 6.0 |
| | Eastern | Vodla | 4 | Min. | −12.2 | −92 | 5.6 | −11.3 | −85 | 5.4 |
| | | | | Max. | −11.0 | −89 | −1.0 | −9.7 | −78 | −0.4 |
| | Southeastern | Andoma | 3 | Min. | −13.1 | −100 | 4.8 | −10.6 | −80 | 4.8 |
| | | | | Max. | −12.5 | −100 | 0.0 | | | |
| | Southwest | Derevyanka | 4 | Min. | −14.6 | −107 | 9.8 | −11.7 | −83 | 10.6 |
| | | | | Max. | −13.1 | −104 | 0.8 | −11.3 | −86 | 4.4 |
| | | Sheltozerka | 2 | Min. | − | − | | −10.8 | −83 | 3,4 |
| | | | | Max. | − | — | | −10.2 | −77 | 4,6 |
| The central part of the East European Platform (EEP) (II) | Southern | Vytegra | 4 | Min. | −12.5 | −97 | 3.0 | −11.7 | −88 | 5.6 |
| | | | | Max. | −11.8 | −90 | 4.4 | −11.1 | −87 | 1.8 |
| | | Svir' (outflow from lake) | 5 | Min. | −10.9 | −78 | 9.2 | −9.7 | −75 | 2.6 |
| | | | | Max. | −10.3 | −81 | 1.4 | −9.6 | −73 | 3.8 |

n—Number of measurements; «–»—no data.

It can also be noted that the isotopic composition the water of the Shuya, Suna and Sheltozerka rivers, regardless of the season, was close to the isotope composition of Lake Onego. Firstly, this increase in the abundance of heavy isotopes in the water of the above-named tributaries was the result of the wide distribution of the lacustrine and swampy landscape on its watershed. Secondly, it confirms the main contribution of the river runoff to the lake's water balance. The dispersion area of the figurative points for the composition of Lake Onego's water on the isotopic diagram occupies a narrow range, which is located in the heavier values of $\delta^2$H and $\delta^{18}$O, and it wasthe most evaporated water among the studied objects. So, it confirms a well-known pattern: the concentration of heavy isotopes in the water of continental lakes is higher than in tributaries [33].

According to the data in the literature, the bulk of the microcomponents is transported by river waters as part of a suspended substance [34]. However, in our opinion, the importance of the dissolved forms should not be underestimated, especially in conditions of a humid climate in the taiga zone, as they play an important role in the formation of continental runoff and are one of the factors in the formation of the ecological and geochemical backgrounds of water systems. The fluctuations in the concentrations of dissolved metals in the tributaries of Lake Onego during the observation period were large: Cr 0.1–1.5 µg/L; Mn 0.43–124 µg/L; Fe 35–1610 µg/L; Co 0.01–0.31 µg/L; Ni 0.22–2.45 µg/L; Cu 0.43–13.6 µg/L; Zn 0.1–13.28 µg/L; As 0.13–0.88 µg/L; Mo 0.01–0.40 µg/L; Cd 0.00–0.04 µg/L; Sb 0.01–0.09 µg/L; Ba 0.51–29.46 µg/L; and Pb 0.02–4.91 µg/L. The average concentrations of most of the dissolved forms of the elements did not significantly differ from those in the world river flow [35] and in the rivers of the catchments of the White and Kara Seas [36] (Figure 3). Discrepancies compared with the values of the world flow were obtained for Fe, Zn and Pb. High concentrations of these elements may be a reflection of regional peculiarities together with anthropogenic influence [2].

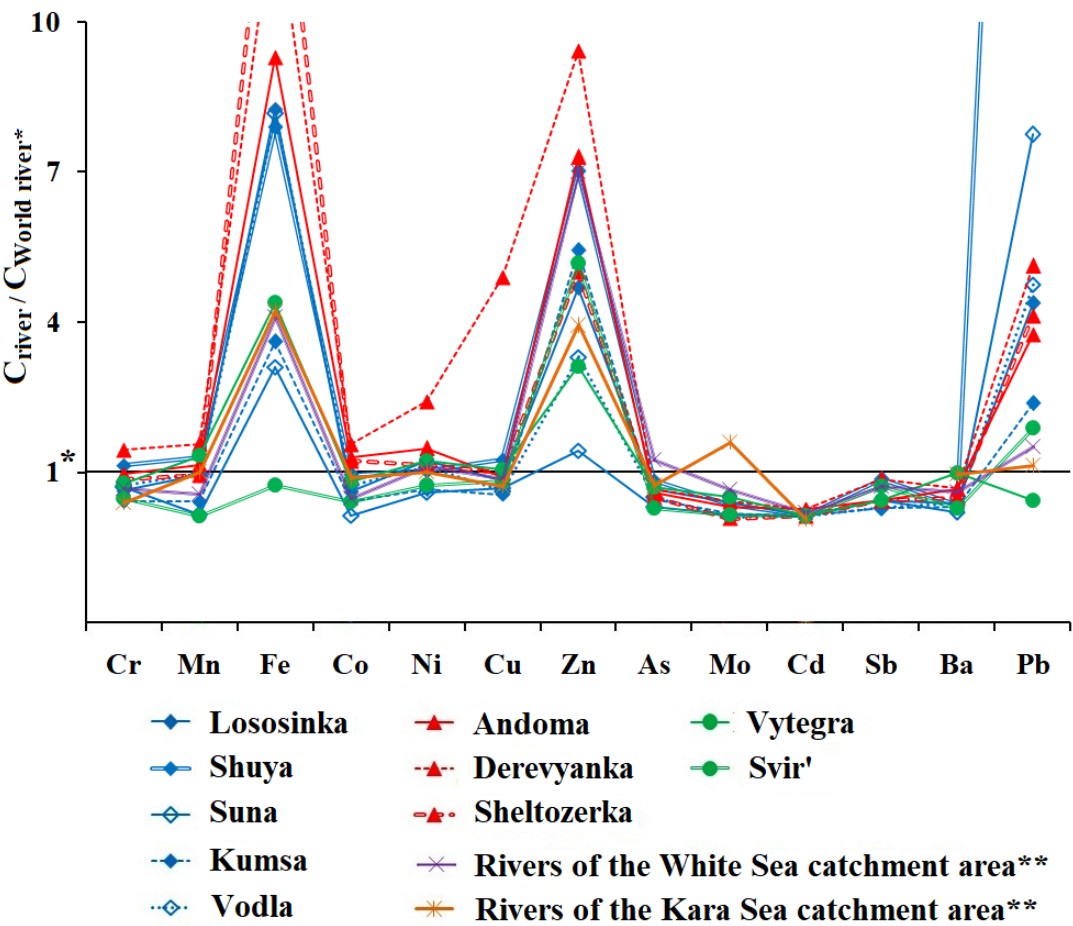

**Figure 3.** The content of dissolved metals in the waters of the studied rivers in relation to the rivers of the world (*). The arithmetic mean of the elemental content of dissolved metals in the river waters was used for the calculation according to the data in the literature. * [35]; ** [36].

### 3.2. Suspended Matter in River Waters

According to published data, 80–90 thousand tons of suspended matter enter Lake Onego annually, which does not exceed 10% of the total amount of matter brought into the reservoir [14]. The main supplier of suspended matter to Lake Onego are rivers, the mineral part of the material of which mainly consists of quartz and feldspar [37].

In the river water samples studied by us, a predominance of suspended matter particles larger than 0.8 μm was observed (Table 5). The content of coarse suspended matter (Ø > 0.8 μm) for all of the studied watercourses ranged from 0.7 to 29.6 (on average 6.1 mg/L) and for fine (0.45 μm < Ø < 0.8 μm) from 0.03 to 9.4 (on average 1.02 mg/L). For rivers draining the territory of the FCS, with the exception of the Derevyanka, Sheltozerka and Andoma rivers, the average annual concentration of suspended matter (coarse—5.5 and fine—0.8 mg/L) was lower than for the Vytegra River (10.4 and 0.6 mg/L, respectively) draining the territory of the central part of the East European Platform. The rivers Derevyanka, Sheltozerka and Andoma, whose catchments are located on the border of the FCS, are characterized by a comparable content of coarse suspended matter (on average 7.4 mg/L) and a high content of fine suspended matter in relation to all of the studied watercourses (on average 1.9 mg/L) (Table 5).

**Table 5.** Seasonal variations of suspended matter content in the surface water layer of the tributaries of Lake Onego.

| Geological Structure | Coast | River | Season | Suspended Matter, mg/L | | |
|---|---|---|---|---|---|---|
| | | | | > 0.8 μm | 0.45 μm < Ø < 0.8 μm | > 0.45 μm |
| The Fennoscandian Crystalline Shield (FCS) (I) | Northwest | Lososinka (before the city) | Spring | 2.96 | 1.80 | 12.26 |
| | | | Summer | 6.25 | 0.48 | 11.67 |
| | | | Autumn | 5.22 | 0.13 | – |
| | | | Winter | 29.59 | 1.06 | 32.80 |
| | | Lososinka (the mouth of the river) | Spring | 3.49 | 2.50 | 5.37 |
| | | | Summer | 7.60 | 0.89 | 1.88 |
| | | | Autumn | 8.12 | 0.04 | – |
| | | | Winter | 8.65 | 0.29 | 9.82 |
| | | Shuya | Spring | 7.36 | 0.75 | 7.12 |
| | | | Summer | 9.82 | 0.59 | 4.47 |
| | | | Autumn | 4.19 | 0.13 | 5.56 |
| | | | Winter | 7.88 | 1.07 | 9.81 |
| | | Suna | Spring | 1.20 | 0.51 | 1.87 |
| | | | Summer | 1.99 | 0.22 | 2.22 |
| | | | Autumn | 0.87 | 0.03 | – |
| | | | Winter | 0.72 | 0.23 | 0.88 |
| | North | Kumsa | Spring | 1.50 | 0.37 | – |
| | | | Summer | 2.35 | 0.15 | 1.96 |
| | | | Autumn | 0.93 | 0.12 | 1.18 |
| | | | Winter | 2.11 | 0.29 | 2.60 |
| | Eastern | Vodla | Spring | 7.76 | 1.35 | 3.72 |
| | | | Summer | 3.24 | 0.22 | 2.20 |
| | | | Autumn | 3.93 | 0.24 | 2.29 |
| | | | Winter | 4.22 | 5.94 | 0.98 |
| | Southeastern | Andoma | Spring | 5.51 | 1.54 | 9.62 |
| | | | Summer | 9.16 | 0.60 | 8.10 |
| | | | Autumn | 3.53 | 0.63 | 4.40 |
| | | | Winter | 9.19 | 1.90 | 12.92 |
| | Southwest | Derevyanka | Spring | 4.69 | 1.79 | 9.13 |
| | | | Summer | 4.83 | 1.10 | 14.17 |
| | | | Autumn | 4.17 | 9.40 | – |
| | | | Winter | 13.00 | 1.12 | 25.47 |
| | | Sheltozerka | Spring | 7.88 | 1.91 | 21.11 |
| | | | Summer | 1.79 | 1.12 | 14.68 |
| | | | Autumn | 2.08 | 0.23 | 5.60 |
| | | | Winter | 22.77 | 1.40 | 16.59 |
| The central part of theEast European Platform (II) | Southern | Vytegra | Spring | 8.29 | 0.90 | – |
| | | | Summer | 21.46 | 0.33 | – |
| | | | Autumn | 5.80 | 0.30 | 4.53 |
| | | | Winter | 6.15 | 0.70 | 6.75 |
| | | Svir' (the source of the lake) | Spring | 2.15 | 0.28 | – |
| | | | Summer | 0.80 | 0.08 | – |
| | | | Autumn | 3.39 | 0.05 | – |
| | | | Winter | 1.07 | 0.11 | 0.96 |

«–»—No data.

At the source of the lake, the content of coarse suspended matter rangedfrom 0.8 to 2.2 (on average 1.9) and forfinefrom 0.05 to 0.3 (on average 0.1 mg/L) (Table 5). Unfortunately, the data obtained do not allow us to clearly identify the period of the maximum intake of suspended matter into the lake. For most rivers, the maximum fell in winter and for some in summer (Suna and Vytegra) and spring (Vodla and Svir'). The minimum content

of the fine fraction of suspended matter in the rivers fell in the autumn period, with the exception of the Vodla, Andoma and Derevyanka rivers, where it was recorded not only in the autumn and summer (Table 5).

### 3.3. Content of Metals in Suspended Matter from the Water of the Tributaries of Lake Onego

The information obtained on the trace element composition of the suspended matter is limited due to the fact of its small amount allocated by filtering the river waters through membrane filters. The turbidity of the rivers, which is an indicator of the content of suspended matter in the water, changed very significantly during the observations. At the same time, the concentration ratios of the elements in the composition of the suspended matter were generally preserved. The accumulation of chemical elements in suspended matter, according to the published literature, is determined by the degree of the individual solubility and the sorption of trace elements [38].

The results of the content of trace elements in the suspended matter of the water of the tributaries of Lake Onego in terms of dry matter are characterized by an increased content of Mn, Fe, Cu, Mo, Cd, Sb and Pb. The remaining metals are at a comparable level with the elemental content in the suspended matter of the waters of the rivers of the world [39–43] (Figure 4).

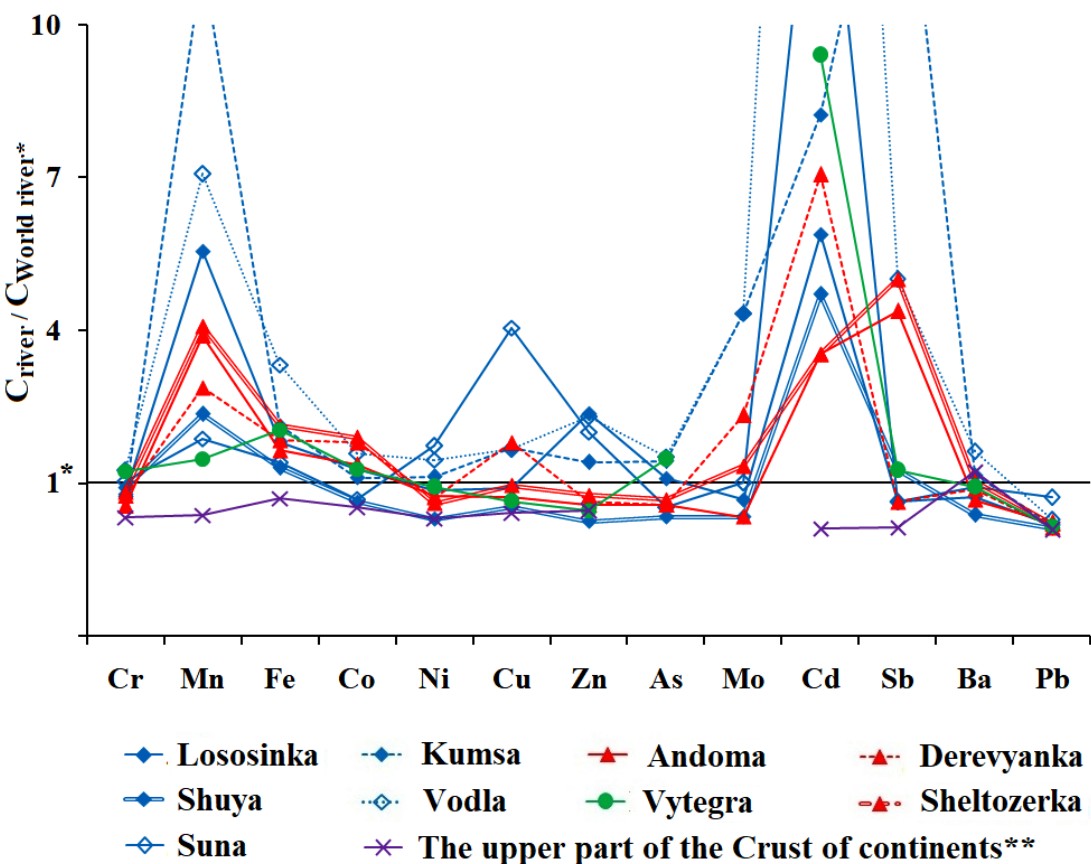

**Figure 4.** The average content of metals in the suspended matter of the water of the surface horizon of the tributaries of Lake Onego in terms of g of dry matter in relation to the content of metals in the suspended matter of the rivers of the world according to the data in the literature. * [39–44]; ** [45].

**Cr:** The average annual Cr content in the suspended matter of the tributaries of Lake Onego was 94 µg/g, which approximately corresponds to the global average Cr content in river waters (Figure 4). The maximum content (210 µg/g) was obtained in the suspended matter of the Vodla River in winter. However, it was not possible to identify the seasonal dependence of the Cr content in the suspended matter of all rivers in our study. Separately,

we would like to note the results obtained on the Cr content in the suspended matter of the Lososinka River. We took water samples and isolated the suspended matter from them at two stations along the river: before the City of Petrozavodsk's territory and at the mouth of the river in the territory of the city. Despite the pronounced anthropogenic influence of the city territory on the river, there were no differences in the level of Cr content in the suspended matter. At the same time, seasonal differences were obtained for the maximum concentration, namely, for the station located above the city line, the maximum concentration of Cr was observed in the winter hydrological season and for the station at the mouth of the riverin the summer.

**Mn:** The concentration of suspended Mn in the water of the tributaries of Lake Onego fluctuated during the year from 649 to 30,281 μg/g (the average value in terms of dry matter was 7401 μg/g). Rivers are characterized by the high variability of Mn content both in water [16] and in suspended matter, which is probably due to the geological features and diversity of the soil cover of their catchments. According to published data [46], a significant amount of Mn accumulates in the forest litter of the region and, in some sample areas, significantly (4–5 times) exceeds the average values for Karelia, amounting to 1.5–3 MPC in the soil. There was a significantly lower amount of Mn in the mineral horizons than in the organogenic ones. In the Zaonezhye and Prionezhsky districts, the accumulation of Mn exceeded the average values for Karelia, but did not reach the MPC in soils. The results obtained on the suspended Mn content in water of the studied rivers were five times higher than the global average (Figure 4) and seven times higher than the Mn content in the upper part of the continental crust (Figure 4). The maximum concentrations of suspended Mn in most rivers were obtained in the summer hydrological season, with the exception of the Suna (in winter), Kumsa and Vytegra (in autumn) rivers.

**Fe:** The geochemical features of Karelian landscapes are a high degree of swampiness of the territory and an increased Fe content in soils. This element enters water bodies from the catchment area in combination with humus substance [47] and is a sign of the Fe-Mn of the province [48]. The values of the concentration of Fe in the suspended matter of the tributaries of Lake Onego ranged from 33 to 345 μg/mg (the average value in terms of dry matter was 97.5 μg/mg), which is two times higher than the global average (Figure 4). The maximum concentration of Fe was observed in the winter for the rivers Lososinka, Shuya, Suna and Andoma and in autumn in the other tributaries of Lake Onego.

**Co:** The average content of suspended Co in the water of the tributaries of Lake Onego for four hydrological seasons in terms of dry matter was 25 μg/g (the concentrations ranged from 4 to 56 μg/g). The obtained values are 1.3 times higher than the global average and exceed the values for the upper part of the continents (Figure 4), the average values correspond to the obtained values of Co in the suspended matter of the rivers of southern France [49], and the limits of fluctuations are similar to the data obtained for different sections of the Amazon River [50]. With the exception of the Kumsa, Vodla and Sheltozerka rivers, the maximum concentration of suspended Co was observed in the winter season for the rivers draining the territory of the FCS. For the Vytegra, Sheltozerka, Kumsa and Vodla rivers, the maximums of the suspended Co content were obtained in autumn.

**Ni:** The content of suspended Ni in the water of the studied rivers in terms of dry matter (on average 59 μg/g with a total spread from 2 to 191 μg/g) did not exceed world data [39–43]. The maximum concentration of Ni in the suspended matter ofthe rivers draining the territory of the FCS (with the exception of the Kumsa river) was obtained in the winter.

**Cu:** According to the data in the literature, the proportion of the suspended form of Cu increases due to the receipt of terrigenous and biological material from the catchment area of rivers and depends on the season [51]. The total spread of the values of the Cu content in the suspended matter of the tributaries of Lake Onego in our study ranged from 0 to 478 μg/g in terms of dry matter. The average value of 87 μg/g is 1.5 times higher than the world data (Figure 4). The maximum concentrations of suspended Cu for most rivers were obtained in the winter hydrological season and for the Andoma and Vodla rivers in the autumn. The results obtained are consistent with previous studies [15,51], and the

increased Cu content in the suspended matter of the tributaries and lake water indicates the possibility of the formation of a geochemical anomaly in the lake.

**Zn:** The Zn content in the suspended matter of the rivers was characterized by a large range of values. According to the data in the literature, with high suspended Zn contents of hundreds of μg/g, during large spring floods, its concentration can decrease by almost an order of magnitude due to the fact of dilution. The data obtained by us for the tributaries of Lake Onego show that changes in the Zn content in the suspended matter (from 0 to 1032 μg/g; on average 189 μg/g) do not significantly exceed the global average (by 1.2 times) and its content in the upper part of the crust of the continents (Figure 4).

**As:** The concentration of As in the rivers' suspended matter over the year varied from 0 to 49 μg/g (average value of 19 μg/g), which generally corresponds to its content in the rivers of the world (Figure 4). With the exception of the Lososinka River, the maximum suspended As in all rivers was obtained during the winter hydrological season. A comparison of the results obtained from two stations along the Lososinka River did not show significant differences (before and within the City of Petrozavodsk), which indicates a natural source of its intake from the catchment area.

**Mo:** The wide limits of the suspended Mo content in the world's water courses are due to the diversity of rocks in catchment areas and variations of organic substances in the composition of suspended matter [42]. In our study, the spread of the suspended Mo content in the tributaries of Lake Onego ranged from 0 to 28 μg/g (average value of 5 μg/g), which is 1.7 times higher than the global values (Figure 4).

**Cd:** The concentration of Cd in the suspended matter of the tributaries of Lake Onego averaged 8 μg/g (a range of values from 0 to 52 μg/g), which is comparable to studies of the suspended matter of French rivers [52]. The data obtained in our study on the content of suspended Cd were nine times higher than the global values (Figure 4). The maximum contents for most rivers were observed in the winter hydrological season, with the exception of the Kumsa River (summer). According to the data in the literature, Cd migrates mainly in the dissolved form in weakly mineralized river waters, but it is also capable of forming complexes with organic substances. The intake, content and form of the migration of the element is influenced by the catchment area and geological and climatic conditions. The Cd content in soils increases significantly in areas of sulfide deposits, which are characteristic of the Karelo–Kola region [53].

**Sb:** The contents of suspended Sb in the river water studied by us were characterized by a large range of values, from 0 to 70 μg/g (average value of 6 μg/g), which is four times higher than the average value for the rivers of the world (Figure 4). For most tributaries of Lake Onego, the maximum concentration was obtained in the autumn hydrological season, with the exception of the rivers Lososinka and Suna (summer).

**Ba:** The average value of the Ba content obtained in the suspended matter of the tributaries of Lake Onego was 403 μg/g, with a range of values from 57 to 1254 μg/g, which is comparable with the data on the rivers of the world (Figure 4). According to the literature, the migration of Ba is mainly associated with the migration of Fe complexes with organic matter [54].

**Pb:** The results obtained in our study on the content of suspended Pb in river water, which ranged from 9 to 235 μg/g (average value of 60 μg/g), are in good agreement with the data on the rivers of the world (Figure 4). According to the literature, the predominant form of Pb migration is the thinnest terrigenous suspended matter (99.6% of all Pb, being in the water) [55].

Based on the research data for each of the tributaries of Lake Onego, series of elements were constructed to reduce therole of a weighted form (Table 6).

**Table 6.** Series of elements to reduce therole of the weighted form in the tributaries of Lake Onego.

| Geological Structure | Coast | River | Series of Elements |
|---|---|---|---|
| I | Northwest | Lososinka | Fe > Mn > Zn > Ba > Cr > Ni > Cu > Pb > Co > As > Cd > Mo > Sb |
| | | Shuya | Fe > Mn > Ba > Cr > Zn > Cu = Pb > Ni > Co > As > Cd > Sb > Mo |
| | | Suna | Fe > Mn > Ba > Zn > Cu > Pb > Ni > Cr > Co = Cd > As > Mo > Sb |
| | North | Kumsa | Fe > Mn > Ba > Zn > Cu > Cr > Ni > Pb > As > Sb > Co > Mo > Cd |
| | Eastern | Vodla | Fe > Mn > Ba > Zn > Cr > Cu > Ni > Pb > As > Co > Cd > Mo > Sb |
| | Southeastern | Andoma | Fe > Mn > Ba > Zn > Cr = Pb > Ni > Cu > Co > As > Sb > Cd > Mo |
| | Southwest | Derevyanka | Fe > Mn > Ba > Cu > Zn > Cr > Ni > Co = Pb > As > Mo > Cd > Sb |
| | | Sheltozerka | Fe > Mn > Ba > Zn > Cr > Cu > Pb > Ni > Co > As > Sb > Mo > Cd |
| II | Southern | Vytegra | Fe > Mn > Ba > Cr > Zn > Ni > Cu > Pb > As > Co > Cd > Sb > Mo |

*3.4. Mineral Composition of Suspended Matter of Rivers*

Electron microscopic examination (SEM) of the samples of the suspended matter of the rivers isolated on filters showed that the concentration level of the mineral component of the sediment corresponds to the data in the literature for the rivers of the Arctic territories [55,56]. The spectra of the minerals of the suspended matter of the tributaries of Lake Onego were close. The quantitative ratios of the mineral and organic parts of the solid suspended matter, the ratios of the different minerals and the size and patterning of the particles of the detrital material in the studied samples differed (Figures 5–7). We established that the suspended matter material of the rivers is represented by a biogenic X-ray amorphous mass (biodetrite of diatoms, spores and pollen of plant communities) with associations of detrital mineral particles, scaly formations of layered silicates and aluminosilicates, and jelly-like clots and films of fouling on the organic skeletons of oxides/hydroxides of manganese (Figures 7 and 8). Among the biogenic particles, plant fibers dominated in the suspended matter of the Vytegra, Vodla and Kumsa rivers (Figure 8). It is important to note that there were large individual grains of minerals (more than 20–50 microns), but mostly they were aggregates of isometric shape, consisting of molded, small grains of minerals (pelitic dimension) and biodendrite particles (small fragments of diatoms and spherical particles of pollen and spores) into pellet lumps (Figures 7A and 9A).

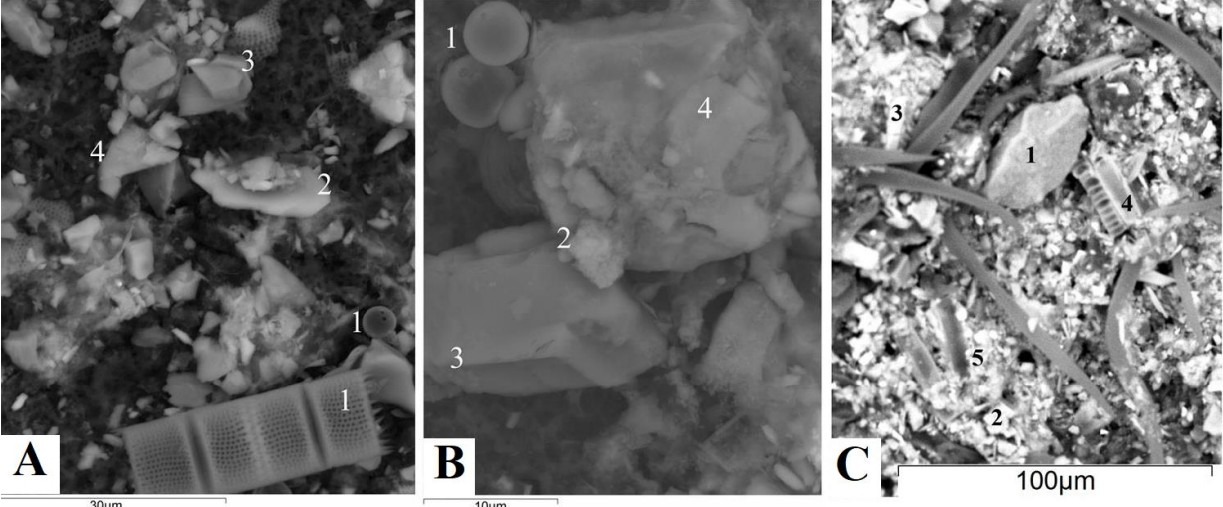

**Figure 5.** Micrographs made using SEM of samples of river suspended matter on filters taken in the Shuya (**A**), Vodla (**B**) and Sheltozerka (**C**) rivers: (**A**) 1—shells and biodetrite of diatoms, 2—muscovite, 3—quartz and 4—K-feldspar; (**B**) 1—shells and biodetrite of diatoms, 2—Fe-Mn oxides and hydroxides, 3—plagioclase (10–30% anorthite) and 4—quartz; (**C**) 1—quartz, 2—K-feldspar, 3—epidote, 4—shells and biodetrite of diatoms and 5—Mg-Fe illite.

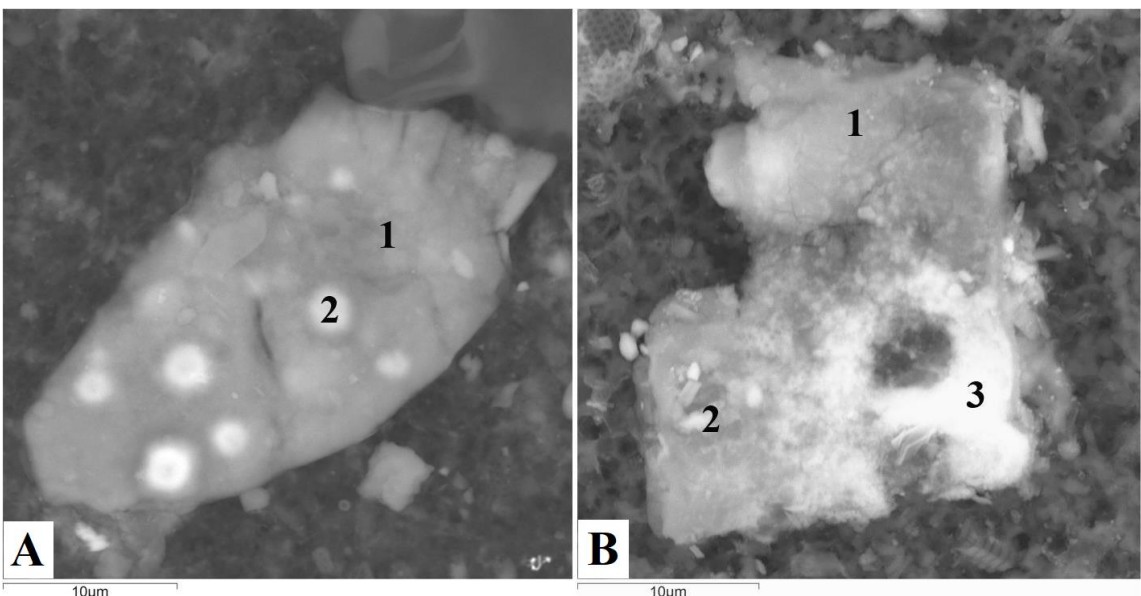

**Figure 6.** Micrographs made using SEM of samples of river suspended matter on filters taken in the Andoma River (**A**) and the Lososinka River (**B**): (**A**) 1—coarse-flaked Mg-Fe chlorite aggregate and 2—latent-flaked ferruginous chlorite grains; (**B**) 1—fine-flaked Mg-Fe illite aggregate, 2—latent scaly grains of Fe illite and 3—a fine-fiber aggregate of Mn oxides.

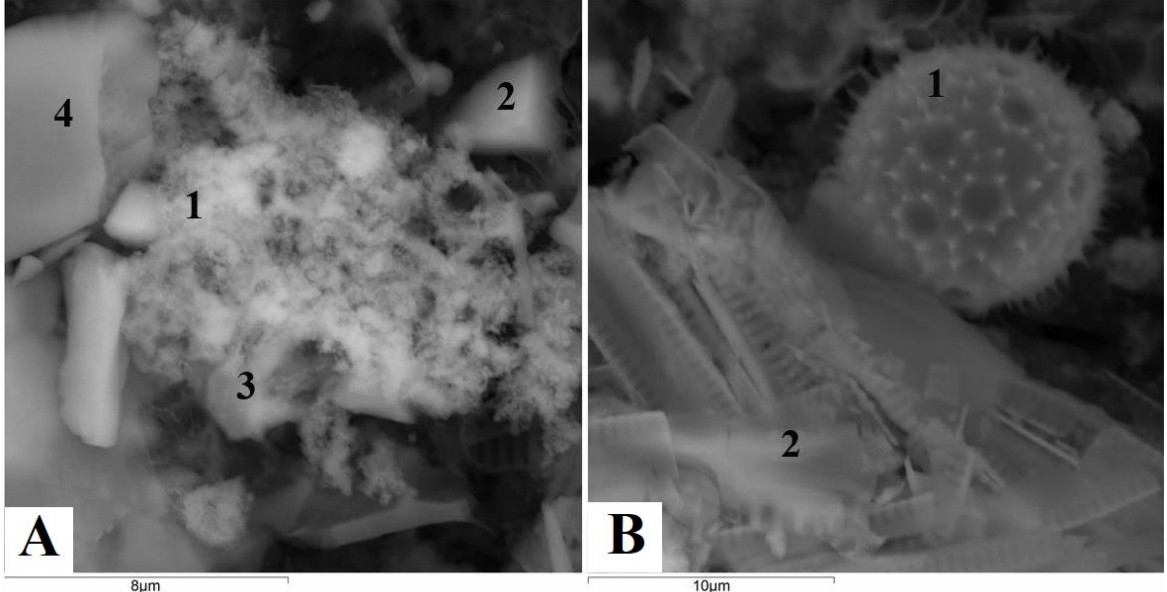

**Figure 7.** SEM photos: (**A**) suspended matter of the Shuya River filter: 1—earthy aggregates of rhodochrosite, 2—quartz, 3—shells and biodetrite of diatoms and 4—plagioclase (10–30% anorthite); (**B**) suspended matter of the Kumsa River filter: 1—pollen and 2—numerous species of diatoms.

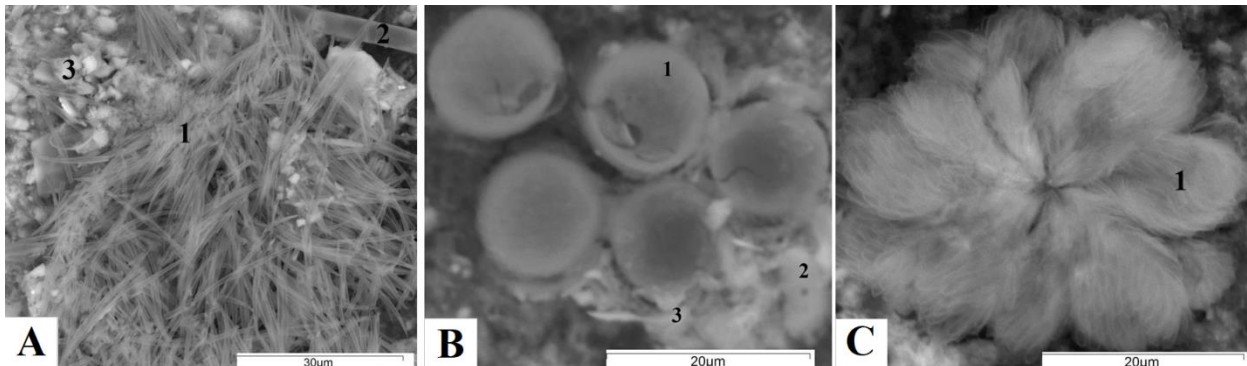

**Figure 8.** Micrographs obtained using SEM ofsamples of river suspended matter on filters taken in the Vytegra River (**A**), Vodla River (**B**) and Kumsa River (**C**): (**A**) 1—fine-fiber algae aggregate (amorphous silica), 2—shells and biodetrite of diatoms and 3—fine-grained aggregate terrigenous minerals (quartz, plagioclase (10–30% anorthite), albite and Fe-Mn illite); (**B**) 1—spherical hollow aggregates of spores, 2—shells and biodetrite of diatoms and 3—scaly aggregates of muscovite; (**C**) 1—fine-fibrous aggregate of algae (amorphous silica).

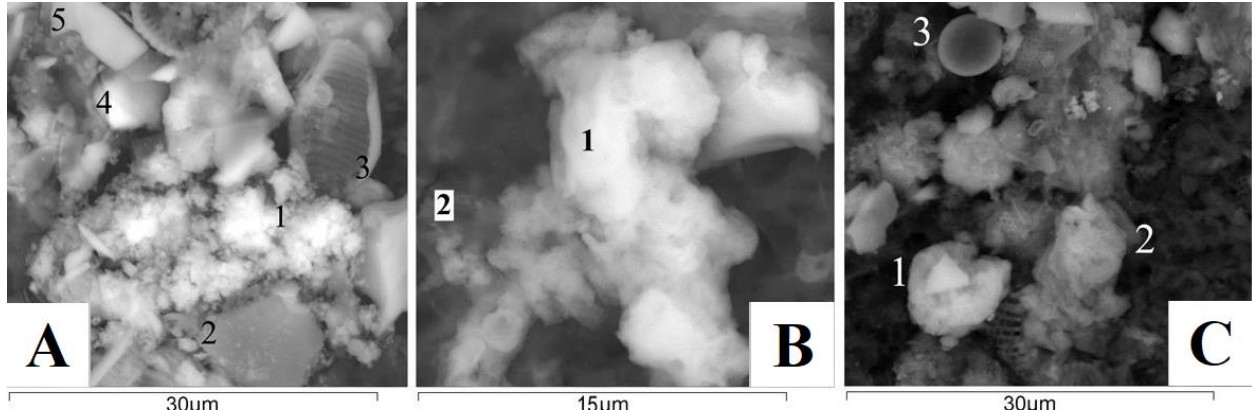

**Figure 9.** Micrographs made using SEM of the aggregates of manganese minerals in river suspended matter on filters taken in the Sheltozerka River (**A**), Vytegra River (**B**) and Lososinka River (**C**): (**A**) 1—earthy aggregates of rhodochrosite, 2—quartz, 3—shells and biodetrite of diatoms, 4—Fe-Mn illite and 5—albite; (**B**) 1,2—sintering aggregates of Mn oxides and hydroxides of varying degrees of crystallinity; (**C**) 1—sintering aggregates of Mn oxides and hydroxides with an admixture of Ba (up to 4%), 2—earthy aggregates of Fe and Mn oxides and hydroxides with an admixture of Ba (1%) and P (1%) and 3—shells and biodetrite of diatoms.

In all samples of suspended matter, individual grains or crystals of dark-colored and accessory minerals, such as epidote, actinolite, hornblende, augite, magnetite, ilmenite, rutile, titanite, apatite andmonazite, were found in very small quantities. The mineral composition of river suspended matter slightly depends on the season of the year. Suspended matter from various tributaries differedin the absolute content of sediment on the filter, particle size, quantitative ratio of basic minerals, presence (absence) of grains of unusual composition and morphology of aggregates of iron minerals. Jelly-like aggregates of iron and manganese oxides/hydroxides of varying degrees of crystallinity and sizes were present in the mineral matter of all rivers in significant quantities. Their greatest concentrations were found in the suspended solids of the Lososinka, Vytegra and Sheltozerka rivers (Figures 7 and 9). In a number of the rivers, in addition to jelly-like iron aggregates, there was a significant number of iron aggregates of unusual morphology: oolites (spherical aggregates), complex cellular aggregates and thin-fiber radially radiant aggregates (Figure 10). Anthropogenic particles in river suspended matter are extremely

poorly represented. Basically, these are individual, small (from 2 to 25 microns) and irregularly shaped grains with a chemical composition that does not occur in nature. The largest number of technogenic particles was recorded in the suspended matter of the Suna and Andoma rivers. For example, in the suspended matter of the Suna River in the composition of terrigenous material, in association with small grains of quartz, calcite and albite with biodetrite of diatoms, there were large and small grains of irregular shape of the alloy Zn-Ni-Cu and Ni-Sn, with a sharp predominance in the composition of Ni (Figure 11A,B). In addition, technogenic particles of native antimony and silver sulfate were found in the suspended matter of the Andoma River (Figure 11C,D).

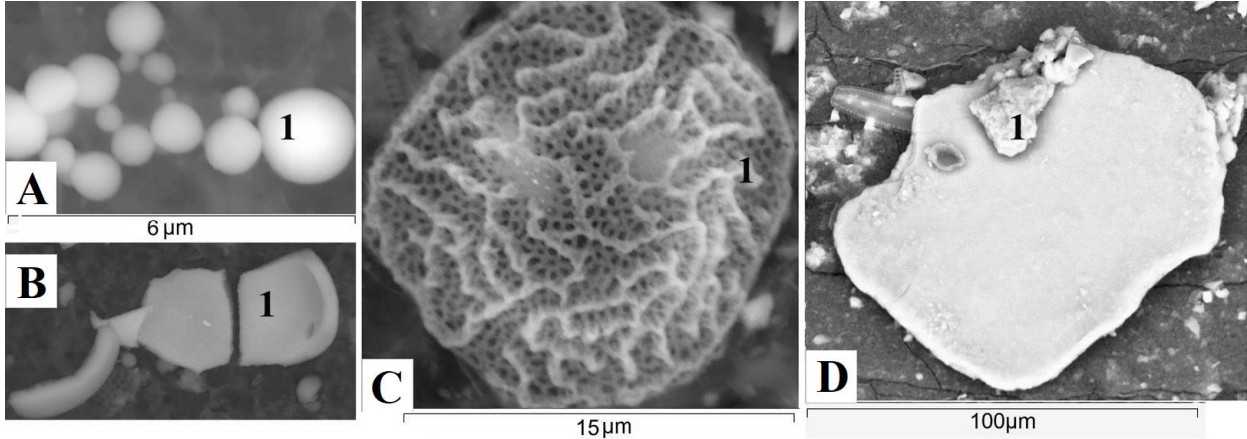

**Figure 10.** Micrographs made using SEM of aggregates of iron minerals of various morphologies in river suspensions on filters taken in the rivers Vodla (**A**), Andoma (**B**), Shuya (**C**) and Sheltozerka (**D**): (**A**) 1—spherical aggregates of goethite; (**B**) 1—spherical fragments of goethite aggregate; (**C**) 1—goethite cellular aggregate; (**D**) 1—large particle of organic matter covered with a film of native iron.

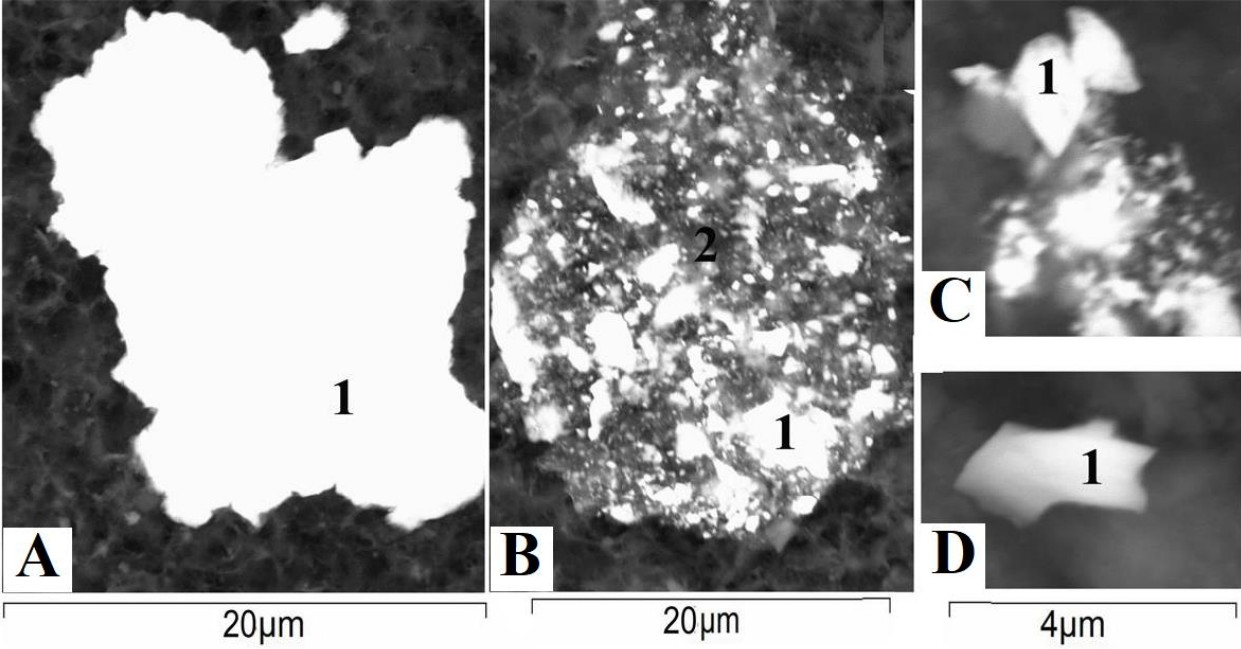

**Figure 11.** Micrographs made using SEM of aggregates of technogenic composition of various morphologies in river suspended matter on filters taken in the Suna (**A,B**) and Andoma (**C,D**) rivers: (**A**) 1—Zn-Ni-Cu alloy aggregate; (**B**) 1—Ni-Sn alloy aggregate and 2—small grains of calcite in association with biodetrite of diatoms; (**C**) 1—crystals and isometric grains of silver sulfate; (**D**) 1—coalescence of grains of native antimony.

The data obtained for the first time on the mineralogy and geochemistry of the suspended matter of the tributaries of Lake Onego were compared with the composition of the suspended matter of the water column of the lake area into which the corresponding river flows: Gulf of Povenets—Kumsa River; Kondopoga Bay—Suna River; Petrozavodsk Bay—Shuya, Derevyanka and Lososinka rivers; Central Onego—Vodla and Sheltozerka rivers; Southern Onego—Andoma and Vytegra rivers and the source of the lake, Svir' River (Figure 1). Fine micro- and nanoparticles of suspended matter entering the lake are aggregated either under the influence of biota, with the transformation of suspended matter into larger amorphous lumps, or due to physicochemical processes (coagulation and flocculation of colloids) [57]. In [37,58,59], it was found that in the composition of the deposited aggregated suspended matter in the water of Lake Onego, in contrast to the suspended matter of its tributaries, fragments of skeletons of diatoms are present in significant quantities. Moreover, among the mineral particles grouped into aggregates ($\varnothing$ 40–100 microns) of suspended matter of the rivers, grains of larger quartz and feldspar predominate. Scaly formations of aluminosilicates were mainly represented by muscovite, illite and chlorite with magnesium and iron content in a ratio of 1:1. The morphology of the aggregates of Fe hydroxides and Mn carbonates was mainly earthy. It should be noted that, in general, the mineral part of the suspended matter entering the lake with rivers corresponds to the suspended matter of the lake waters by geochemical characteristics. The suspended matter from the Gulf of Povenets water was close in material composition to the suspended matter of the Kumsa River. The suspended matter from the waters of the South Onego corresponded to the suspended matter of the Andoma and Vytegra rivers, and, as expected, completely coincided with the composition of the suspended matter in the water of the source from the lake (i.e., Svir' River) (Figure 12). The composition of the suspended matter of the water of the Central Onego logically corresponds to the composition of the suspended matter of the Vodla River and differs significantly from the composition of the suspended matter of the small river of Sheltozerka. The greatest differences are characterized by the composition of the suspended matter of the Suna River from the composition of the suspended matter of the waters of Kondopoga Bay. In Petrozavodsk Bay, a close correspondence was found to the average composition of the suspended matter of the rivers flowing into the bay (Shuya, Lososinka and Derevyanka), with the exception of Na, K, Cr and highly charged elements. For all rivers and the corresponding areas of the lake, when comparing the absolute values of the concentrations of the chemical elements in the sediments isolated on the filters, the largest deviations were noted for Na, K and Cu.

Thus, the geochemical composition of the river waters is in good agreement with the composition of the rocks comprising the catchment areas of the tributaries of Lake Onego. This is the main reason for the differences in the material compositions of the rivers. The material composition and volume of river flow (Table 1) determine the uneven distribution of some elements in the water area of the lake. For example, a higher content of Cr in the suspended matter of the water of the central regions of the lake in relation to other areas is most likely due to the increased concentrations of this element in the suspended matter of the tributaries of the Vodla and Andoma rivers. The relatively high content of Ba in the suspended matter of the Vytegra and Sheltozerka rivers determines the differences in the composition of the suspended matter of the waters of Southern Onego. A significant content of Mn in the suspended matter of the water of the Lososinka River, apparently, may be the cause of the periodically determined high concentrations of this element in the suspended matter of Petrozavodsk Bay. Moreover, this unevenness manifests itself in the distribution of both dissolved and suspended forms of microcomponents over the lake's water area [59].

It should be noted that the material composition of the main tributaries of Lake Onego has not been studied before in such detail. In our work, data on the trace element and mineral composition of suspended matter were obtained for the first time. First, we were faced with the question of to what extent the results obtained in our study reflect reality and are comparable with the published data. In order to control for possible errors during sampling, sample preparation and analysis, intra- and interlaboratory control of the results

were used. The next step was to compare our results with the data on the content of metals in the suspended matter of the rivers of the world based on the published literature. As a result, quite large differences were obtained for several elements.

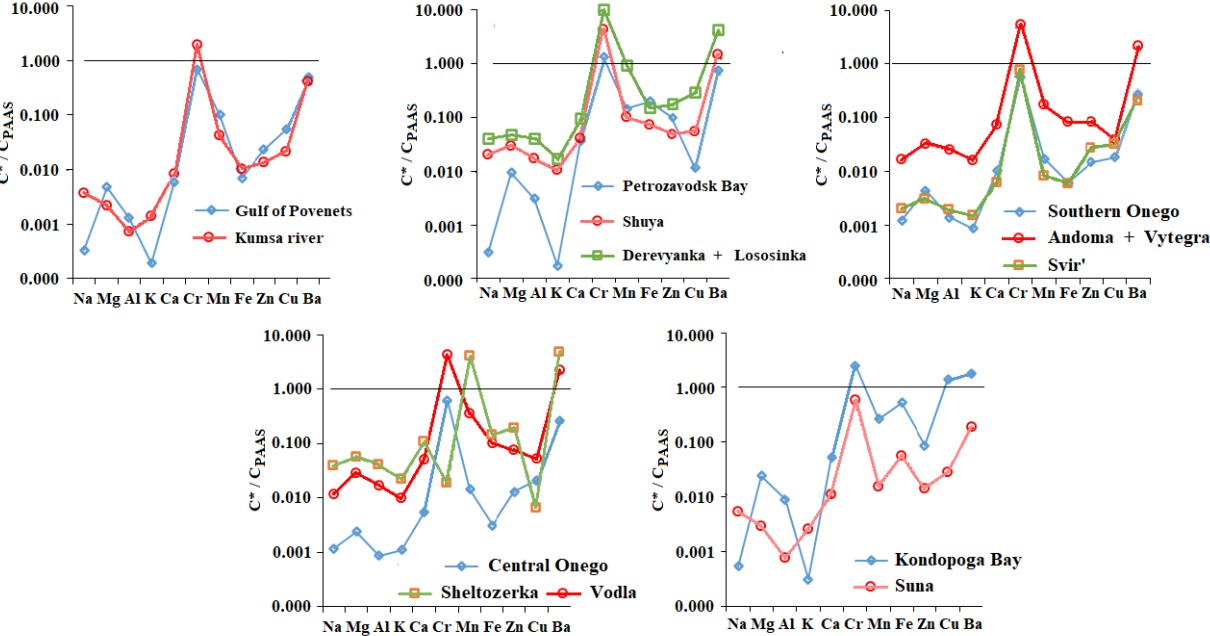

**Figure 12.** The distribution of the content of elements in the suspended matter of water in the estuaries of the rivers and their corresponding areas of Lake Onego, normalized to the content in PAAS [45]. C*—the concentration of the element in the suspended matter.

The increased concentrations of Mn, Fe and Cu in the suspended matter of the rivers were expected, taking into account the specifics of the region, and were confirmed by the published data of researchers in different fields. For elements such as Cr, Co, As, Mo, Cd, Sb and Ba, there is very little or no published data. The results obtained by us were generally comparable with the data on the rivers of the world for Cr, As, Mo, Sb and Ba.

Despite the high variability of the multielement spectrum of the river runoff entering Lake Onego, the lake itself as a whole is characterized by the stability of the chemical composition of the waters. The main reason for the stability of the chemical composition is the ratio of the volume of water coming from the river runoff and the volume of lake waters. The large volume of water of Lake Onego, the long period of water exchange of the lake (16 years), the large depths and the development of cyclonic currents contribute to the dilution of the river waters. River matter (both in the dissolved and suspended form) can be transported over long distances. Stable geochemical conditions of the low-mineralized, oligohumus waters of the calcium bicarbonate class with a neutral pH in the presence of oxygen ensure the transformation processes of humus river waters. The transformation processes are apparently accompanied by the deposition of microelements in the suspended matter, which contributes to the removal of elements from the water column and their accumulation in bottom sediments. These representations correspond to the balance estimates reported in the article by Lozovik P.A. and coauthors [15]. Thus, on the one hand, the waters of Lake Onego inherit the chemical composition of the tributary waters of their main components and, on the other hand, they differ from them by lower concentrations of microcomponents.

Significant inter-seasonal fluctuations in the composition of river waters indicate that the climate is the main factor determining the dynamics of the intake of substances into the lake. In winter, chemical runoff is controlled by the groundwater of river catchments. In spring and autumn, the main source of the substance is the soils of the catchment area. In summer, the contribution of different sources depends on the synoptic situation. The qualitative and

quantitative composition of the suspended forms of elements from the forested watershed of Lake Onego is naturally determined by the composition of the rocks composing the riverbed and floodplain parts of the rivers and the hydrological regime of the rivers.

## 4. Conclusions

River runoff plays a major role in the formation of the geochemical composition of Lake Onego water and affects the direction of the biogeochemical processes of substance transformation in the reservoir. Climatic conditions determine the general characteristics of the chemical composition of river waters. The isotopic composition of the tributary water depends on the season and the characteristics of the catchment area (swampiness andlacustrine). The lightest isotopic composition is characteristic of small rivers, mainly groundwater feeding, and the isotopic composition of large rivers is close to the composition of Lake Onego. The waters of the tributaries of Lake Onego belong to the bicarbonate class of the calcium group and have low mineralization and a relatively high content of dissolved forms of Si, which ensure the development of diatom phytoplankton in the lake, as well as the high color and close composition of the main minerals of the detrital material of suspended matter. The high concentrations of Fe, Zn and Pb in the dissolved form and of Mn, Cu, Cd and Sb in the suspended form obtained in all of the rivers reflect the features of the geochemical province. The concentrations of most metals are at a comparable level with their content in the rivers of the world.

The differences in the material compositions of river waters are primarily due to the heterogeneity of the geological and geomorphological structures of catchments. The waters of the Vytegra River draining the central part of the East European Platform have higher mineralization and higher concentrations of BE. The difference between the multielement spectra of the water and the suspended matter of the different rivers flowing into Lake Onego is closely related to the composition of the rocks of river basins. The ranges of the metal concentrations are significant.

The suspended matter of rivers is represented by biogenic X-ray amorphous mass (biodetrite of diatoms, spores and pollen of plant communities) with associations of detrital mineral particles (quartz, feldspar, illite, muscovite andchlorite), scaly formations of layered silicates and aluminosilicates, jelly-like clots and films of fouling on organic skeletons of oxides, Fe and Mn hydroxides. The quantitative characteristics of the mineral and organic parts of the suspended matter, the ratios of different minerals and the size and patterning of particles of detrital material in the tributaries of Lake Onego vary. The mineralogical and geochemical compositions of the suspended matter of each individual river changes little during the year.

The variability of the chemical composition of dissolved forms of elements in the water of rivers (especially small ones) during the year depends on the season. BE and indicators of organic matter logically showed the greatest variability: $P_{tot}$ from 35 to 168 µg/L; $P_{min}$ from 9 to 58 µg/L; $N_{org}$ from 0.19 to 0.90 mgN/L; N-NH$_4$ from 0.01 to 0.22 mgN/L; NO$_3$ from 0.006 to 0.667 mgN/L; COD from 4 to 70 mgO/L; color from 60 to 360° Pt; and BOD from 0.5 to 7.3 mgO/L.

The influence of river runoff on the formation of lake waters is manifested in the chemical composition of the lake waters. The quantitative ratios of the main ions, BE and microcomponents in the lake water mainly correspond to their ratios in the river waters.The mineral part of the dispersed sedimentary matter of the lake in its geochemical characteristics is close to the suspended matter of river waters.

**Author Contributions:** Conceptualization, N.K., N.B. and V.S.; Methodology, N.K., N.B., V.S., N.E. and G.B.; Validation, N.K., N.E. and V.S.; Formal Analysis, N.K., N.B., V.S., N.E. and G.B.; Investigation, N.K., N.B., V.S., N.E., G.B., E.G., V.M. and I.T.; Data Curation, N.K., N.B., V.S., N.E., G.B. and E.G.; Writing—Original Draft Preparation, N.K., N.B., V.S. and N.E., Writing—Review andEditing, N.K., N.B., V.S., N.E. and G.B.; Visualization, N.K., V.S. and V.M.; Supervision, N.B. and V.S.; Project Administration, N.B. and V.S.; Funding Acquisition, V.S. All authors have read and agreed to the published version of the manuscript.

**Funding:** The study was supported within RSF project #18-17-00176, RFBR, grant #19-05-50014, and state assignment to the Northern Water Problems Institute, KarRC RAS, #FMEN-2021-0006. The isotopic composition of the river waters was studied with the support of the state assignment to the Northern Water Problems Institute, KarRC RAS, #FMEN-2021-0003.

**Data Availability Statement:** The datasets related to this article can be found at https://doi.org/10.17632/5ry8bgfktk.2 (accessed on 25 February 2023) https://doi.org/10.17632/47d4yv26gh.1 and https://doi.org/10.17632/wbs7ztd2sv.1 (accessed on 25 February 2023), an open707 source online data repository hosted at Mendeley Data [60–62].

**Conflicts of Interest:** The authors declare no conflict of interest.

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
