# Peer review of "Geochemical Features of River Runoff and Their Effect on the State of the Aquatic Environment of Lake Onego"

_water, doi:10.3390/w15050964_

Round 1

Reviewer 1 Report

The manuscript “Geochemical features of river runoff and their effect on the state of the aquatic environment of Lake Onego presents interesting information about the geochemical and isotopic composition the large tributaries of Lake Onego. These data are suitable to publish in Water. However, the manuscript should be minor corrected.

General comment. The discussion is need in generalization of obtained data.  For example, what types and season features of areas of the lake catchment are defined inferred from obtained data? It will be useful some discussion as chemical and suspend composition of the tributaries can force on ecosystem of Lake Onego. Has changed or not modern chemical status of the tributaries and anthropogenic effect compare to a previous time span?       

Minor comments.

1. In Introduction, it should be added some short information about why the focus of the paper -“The main attention in these works is paid to the biogenic load on the lake” is a new compare to previous studies.

2. The sentence “According to Alekin's classification,..” is need in the reference.

3. The sentence “The catchments of these rivers are characterized by low degrees of lake content.” is not clear, and should be rephrased.  

4. Table 3. Display of symbol “Aorg” should be checked.

5. It is added data about d-excess will be useful.

6. There is Cyrillic word in tab.6.

7. Settlements should be pointed in fig.1  

Author Response

Dear Reviewer,

thank you for your comments, which have helped us to substantially improve the manuscript.

Response to Reviewer 1 Comments

Point 1: The discussion is need in generalization of obtained data.  For example, what types and season features of areas of the lake catchment are defined inferred from obtained data? It will be useful some discussion as chemical and suspend composition of the tributaries can force on ecosystem of Lake Onego. Has changed or not modern chemical status of the tributaries and anthropogenic effect compare to a previous time span?

Response 1: Unfortunately, in the framework of previous studies, the trace element and mineralogical composition of the suspended matter of river waters has not been studied and it is impossible to compare with the data obtained in our study. According to your recommendations, we have added information about the effect of the chemical composition of river waters on the ecosystem of Lake Onego (read more in the attached file).

Point 2: In Introduction, it should be added some short information about why the focus of the paper -“The main attention in these works is paid to the biogenic load on the lake” is a new compare to previous studies.

Response 2: In accordance with your comment, a summary of previous studies has been added to the article, and the reasons for considering the results obtained in our study as new (read more in the attached file).

Point 3: The sentence “According to Alekin's classification,..” is need in the reference

Response 3: According to your recommendation, a reference to the classification of natural waters of Alekin O.A. was added to the text of the article.

27. Alekin, O.A. Fundamentals of Hydrochemistry. Bruevich, S.V. (Ed.); Hydrometeorological Publishing House: Leningrad, USSR, 1970; pp. 120-121.

Point 4: The sentence “The catchments of these rivers are characterized by low degrees of lake content.” is not clear, and should be rephrased.

Response 4: The proposal was deleted because it does not carry information about the source of organic matter of small rivers (soils).

Point 5: Table 3. Display of symbol “Aorg” should be checked

Response 5: Table 3 has been checked and corrected according to your recommendations.

Point 6: It is added data about d-excess will be useful.

Response 6: In accordance with your comment, information about the d-excess was added to the article, Table 4 was changed (read more in the attached file).

Point 7: There is Cyrillic word in tab.6.

Response 7: Table 6 has been checked and corrected according to your recommendations.

 Point 8: Settlements should be pointed in fig.1

Response 8: Figure 1 has been corrected according to your recommendations.

Reviewer 2 Report

The authors have carried out extensive studies to identify the influence of the geochemical composition of the waters of large tributaries of Lake Onego, the second largest reservoir in Europe, on the formation of lake water. The use of a large mass data on the chemical composition of river waters in different climatic seasons, supplemented by the results of isotope studies, as well as geochemical studies of suspended matter, allowed the authors to validate conclusions by demonstration. Considerable attention was paid to the study of the material composition of suspended matter in river waters, its mineral and organic components. Most significantly, data on the mineralogy and geochemistry of suspended matter of the tributaries of Lake Onego were obtained for the first time.

As a remark, it was somewhat strange to compare the content of such elements as Co and Cd in suspended matter with the its obtained values in the suspended matter of the Amazon River and France rivers, while the content of all the elements (including Co and Cd) considered in this paper was compared with its global average values.

Author Response

Dear Reviewer,

thank you for your comments, which have helped us to improve the manuscript.

Response to Reviewer 2 Comments:

Point 1: As a remark, it was somewhat strange to compare the content of such elements as Co and Cd in suspended matter with the its obtained values in the suspended matter of the Amazon River and France rivers, while the content of all the elements (including Co and Cd) considered in this paper was compared with its global average values.

Response 1: Since the data on the trace element composition of the suspended matter of the tributaries of Lake Onego were obtained for the first time, the question arose before us, first of all, to what extent the results obtained in our study reflect reality and are comparable with the published data. In order to control possible errors during sampling, sample preparation and analysis, intra- and inter-laboratory control of the results was used. The next step was to compare our results with the data on the content of metals in the suspended matter of the rivers of the World based on the published literature. As a result, quite large differences were obtained in several elements.

The increased concentrations of Mn, Fe and Cu in the suspended matter of rivers were expected, taking into account the specifics of the region, and were confirmed by published data of researchers in different fields. For elements such as Cr, Co, As, Mo, Cd, Sb and Ba, there is very little or no published data. The results obtained by us were generally comparable with the data on the rivers of the world for Cr, As, Mo, Sb and Ba.

The Cd content values exceeded the concentrations in the suspended matter of the rivers of the world by 9 times. For Co, the excess concentration was insignificant, but we observed a very large variation in values depending on the hydrological season. A search in the published literature allowed us to find several works where the authors noted similar results, namely the study of the rivers of southern France and the Amazon River. Therefore, these works were mentioned in the description of the results obtained for these elements.

Reviewer 3 Report

The author has studied the geochemistry and isotopic composition of suspended matter in river water, and revealed the chemical and isotopic composition of seasonal tributaries and the characteristics of catchment areas (swampiness and lacustrine).

This article is of very high quality and scientific issues are also interesting. I think it can be accepted and published.

In addition, I suggest that Figure 1 can add a location map of the study area in the world. As an international journal, so that readers can quickly locate in the research area

Author Response

Dear Reviewer,

thank you for your comments, which have helped us to improve the manuscript.

Response to Reviewer Comments:

Point 1: In addition, I suggest that Figure 1 can add a location map of the study area in the world. As an international journal, so that readers can quickly locate in the research area

Response 1: Figure 1 has been corrected according to your recommendations.